

# Computation of longwave radiative flux and vertical heating rate with 4A-Flux v1.0 as integral part of the radiative transfer code 4A/OP v1.5

Yoann Tellier[1], Cyril Crevoisier[1], Raymond Armante[1], Jean-Louis Dufresne[1], and Nicolas Meilhac[2]

[1]LMD/IPSL, École Polytechnique, Institut Polytechnique de Paris, ENS, PSL Research University, Sorbonne Université, CNRS, Palaiseau France
[2]FX CONSEIL, École Polytechnique, F 91128, Palaiseau Cedex, France

**Correspondence:** Yoann Tellier (yoann.tellier@lmd.ipsl.fr)

**Abstract.** Based on advanced spectroscopic databases, line-by-line and layer-by-layer radiative transfer codes numerically solve the radiative transfer equation with a very high accuracy. Taking advantage of its pre-calculated optical depth look-up table, the fast and accurate radiative transfer model Automatized Atmospheric Absorption Atlas OPerational (4A/OP) calculates the transmission and radiance spectra for a user defined layered atmospheric model. Here we present a module, called 4A-Flux, developed and implemented into 4A/OP in order to include the calculation of the clear-sky longwave radiative flux profiles and heating rate profiles at a very high spectral resolution. Calculations are performed under the assumption of local thermodynamic equilibrium, plane-parallel atmosphere and specular reflection on the surface. The computation takes advantage of pre-tabulated exponential integral functions that are used instead of a classic angular quadrature. Furthermore, the sublayer variation of the Planck function is implemented to better represent the emission of layers with a high optical depth. Thanks to the implementation of 4A-Flux, 4A/OP model have participated in the Radiative Forcing Model Intercomparison Project (RFMIP-IRF) along with other state-of-the-art radiative transfer models. 4A/OP hemispheric flux profiles are compared to other models over the 1800 representative atmospheric situations of RFMIP, yielding an Outgoing Longwave Radiation (OLR) mean difference between 4A/OP and other models of $-0.148$ W.m$^{-2}$ and a mean standard deviation of $0.218$ W.m$^{-2}$, showing a good agreement between 4A/OP and other models. 4A/OP is applied to the Thermodynamic Initial Guess Retrieval (TIGR) atmospheric database to analyze the response of the OLR and vertical heating rate to several perturbations of temperature or gas concentration. This work shows that 4A/OP with 4A-Flux module can successfully be used to simulate accurate flux and heating rate profiles and provide useful sensitivity studies including sensitivities to minor trace gases such as HFC134a, HCFC22 and CFC113. We also highlight the interest for the modeling community to extend intercomparison between models to comparisons between spectroscopic databases and modelling to improve the confidence in model simulations.

## 1 Introduction

Atmospheric radiative transfer is the main driver of the climate system and plays a central role in many atmospheric processes. Accurate models that calculate radiative variables such as optical depths, transmittance, radiance, radiative fluxes or vertical





heating rates are currently used for multiple purposes. These algorithms are central to General Circulation Models (GCMs) as the energy balance between net shortwave and net longwave radiation fundamentally drives the climate system. Used as

forward models in the retrieval of geophysical parameters, such codes are also needed in the development and exploitation of Earth observation satellite missions. Thus, radiative transfer models participates in the improvement of our understanding of the atmosphere and the climate.

By resolving the individual spectral lines for every absorbing species at every layer of the atmosphere, line-by-line and layer-by-layer radiative transfer codes offer a very accurate way to compute radiative transfer. Being resource intensive, these

line-by-line calculations are not used in GCMs that use instead a radiation parameterization to approximate the radiative transfer. However, thanks to its high accuracy, the line-by-line radiative transfer codes can be used as a reference to improve the performances of the GCMs by improving their radiation parameterization.

Several line-by-line models have recently been participating to numerical experiments within the Radiative Forcing Model Intercomparison Project (RFMIP, see Pincus et al. (2016)), endorsed by the sixth phase of the Coupled Model Intercomparison

Project (CMIP6, see Eyring et al. (2016)) that aimed at characterizing the clear-sky GCM radiation parameterization error. Pincus et al. (2020) have presented the clear-sky instantaneous radiative forcing (IRF) by greenhouse gases (GHG) computed with these six benchmark models among which 4A/OP line-by-line radiative transfer code.

The Automatized Atmospheric Absorption Atlas OPerational (4A/OP) is a fast and accurate line-by-line radiative transfer model particularly efficient in the infrared region of the spectrum. It offers the calculation of the transmittance, the Jacobians

and the radiance from a user defined atmospheric and surface description. The concept of 4A/OP (Scott and Chedin, 1981; Cheruy et al., 1995) relies on a compressed look-up table of optical depths, called the Atlas, that are computed from reference atmospheres using STRANSAC, the complete line-by-line and layer-by-layer model developed at the *Laboratoire de Météorologie Dynamique* (Scott, 1974; Tournier et al., 1995). 4A/OP interpolates the Atlas to compute the transmittance and the radiance at the right layer temperature, pressure levels and for the desired absorber composition and viewing angle.

4A/OP has a long history of validation within the frame of the international radiative transfer community. Numerous intercomparison exercises have taken place, in particular in the framework of the Intercomparison of Transmittance and Radiance Algorithms working groups of the International Radiation Commission (Chédin et al., 1988) and during the ICRCCM (Intercomparison of Radiation Codes in Climate Models) campaigns (Luther et al., 1988). However, these intercomparisons did not include the modeling of OLR and heating rates. The first developments of the computation of OLR and vertical heating rates

using 4A/OP have been performed in the 1990s (Chéruy et al., 1996; Chevallier et al., 1998, 2000). The computation took advantage of the readily available radiances computed by 4A/OP to calculate the radiative quantities by a spectral and an angular integration. At the time, for computation time reasons, the spectral integrations were often performed after a contraction of the radiance spectra which lead to errors on the modeled fluxes. The spectral integration was performed with either a Gaussian quadrature or at a single angle under the diffuse approximation. Here, different integration approaches have been used.

Firstly, the spectral integration is systematically performed at the finest resolution available. And secondly, exponential integral functions applied to optical depth to directly compute the fluxes has been chosen instead of the quadrature of the radiances.



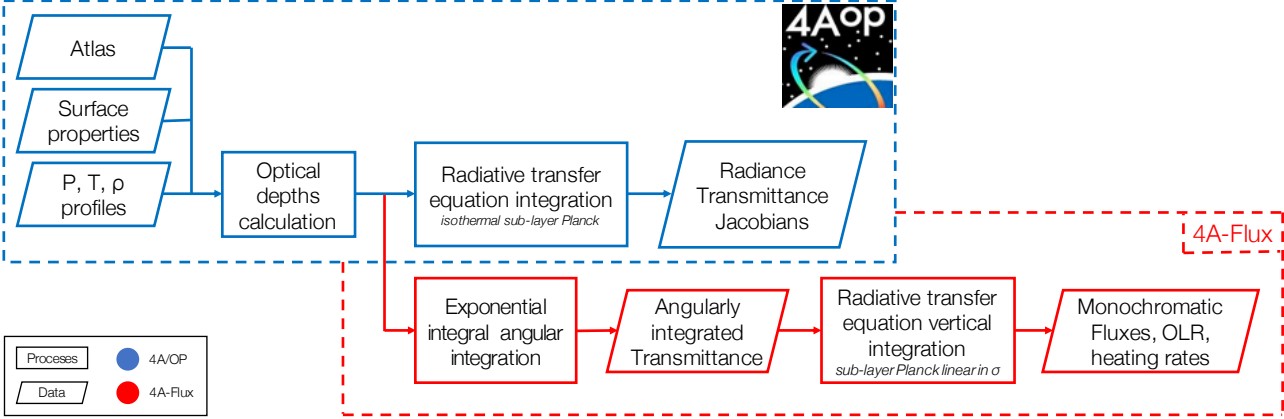

**Figure 1.** 4A/OP and 4A-Flux flowchart. $P$, $T$ and $\rho$ are respectively the pressure grid, the temperature profile and the profiles of gas concentrations.

The main objective of this article is to describe the new implementation of the calculation of the clear-sky longwave radiative flux profiles and vertical cooling rates into 4A/OP radiative transfer code. This module, called 4A-Flux, takes advantage of the efficiency and accuracy of 4A/OP to compute the radiative quantities. Here, we first describe the implementation of 4A-Flux

into 4A/OP. The radiative flux profiles are then compared to the outputs of the five other radiative transfer models that have contributed to RFMIP. Finally, we have applied 4A-Flux module to the computation of the Outgoing Longwave Radiation (OLR) and vertical heating rate sensitivities to several surface and atmospheric parameters.

## 2   Implementation of 4A-Flux into 4A/OP Radiative Transfer Model

### 2.1   General Approach to Flux Computation

4A-Flux module computes the clear-sky longwave downwelling, upwelling and net radiative fluxes (or irradiance) and vertical heating rate at every pressure levels of the user defined atmospheric model and at the finest spectral resolution offered by the line-by-line radiative transfer model 4A/OP of $5 \cdot 10^{-4}$ cm$^{-1}$ (Scott and Chedin, 1981).

Figure 1 presents the flowchart of 4A/OP with its new module 4A-Flux. 4A/OP model provides a suitable basis for the computation of the radiative fluxes as it computes the vertical distribution of total optical depths at the spectral resolution of

absorption lines. The inputs of 4A/OP are the atlases, a user defined description of the pressure grid, the vertical distribution of temperature and gas concentrations along with surface properties such as surface temperature and spectral emittance. In this work, the spectroscopic database used in the calculation of the atlases is GEISA-2015[1] (Jacquinet-Husson et al., 2016) and the continuum used is the MT_CKD 3.2 continuum (Mlawer et al., 2012). The pressure grid, the vertical distribution of temperature and gas concentrations along with surface properties such as surface temperature and spectral emittance can be specified

---

[1]Gestion et Étude des Informations Spectroscopiques Atmosphériques (Management and Study of Spectroscopic Information)





by the user. Based on this description of the atmosphere and optical properties of gases, 4A-Flux computes the upwelling and downwelling monochromatic radiative flux profiles, also referred to as hemispherical monochromatic flux profiles [unit: $W.m^{-2}.(cm^{-1})^{-1}$]. This computation is based on an exponential integral evaluation to perform the angular integration and a vertical integration. In the distributed version of 4A/OP, the vertical integration of the radiative transfer equation is performed under the hypothesis of isothermal sub-layers. Whereas in 4A-Flux, the sub-layer Planck function varies linearly with the

optical depth to better represent the emission of each layer. 4A-Flux also offers the option of computing the monochromatic Outgoing Longwave Radiation (OLR) instead of the full vertical description of fluxes allowing a gain of computational time whenever the user does not require the complete vertical description of fluxes. The output spectral resolution is set by default to $1\ cm^{-1}$ but can be specified by the user at any other value. The net monochromatic flux profile and the monochromatic heating rate vertical profile are then derived from the hemispheric profiles. The following section defines the radiative quantities and

all the assumptions that have been made. We first focus on the definition of the downwelling flux, then we treat the upwelling flux and finally, all the derived quantities that are computed by 4A-Flux are defined.

## 2.2 The Computation of Fluxes and Heating Rate Profiles

The monochromatic[2] hemispheric fluxes [unit: $W.m^{-2}.(cm^{-1})^{-1}$], denoted $F^{\downarrow}(\sigma)$ for the downwelling flux and $F^{\uparrow}(\sigma)$ for the upwelling flux, are represented here by functions of $\sigma$, the monochromatic optical depth [unit: dimensionless] for all

accounted gases in the model, as seen from the top of the atmosphere (TOA), where $\sigma$ is null, to the bottom of the atmosphere (BOA) where it reaches its maximum value $\sigma_{BOA}$. Here, $\sigma$ serves as a vertical coordinate as it increases monotonically with the pressure. The hemispheric fluxes are defined as the integrals of the monochromatic hemispheric radiances $I^{\downarrow}(\sigma,\mu,\phi)$ and $I^{\uparrow}(\sigma,\mu,\phi)$ [unit: $W.m^{-2}.sr^{-1}.(cm^{-1})^{-1}$], over the respective hemispheres (Stamnes et al., 2017; Liou, 2002). These are defined as functions of the vertical coordinate $\sigma$ the azimuthal angle $\phi$ and the direction $\mu = |\cos(\theta)|$ which is the absolute

value of cosine of the zenith angle $\theta$ where $\theta > \pi/2$ for the downwelling flux and $\theta \le \pi/2$ for the upwelling flux:

$$F^{\downarrow\uparrow}(\sigma) = \int\limits_0^{2\pi}\int\limits_0^1 \mu I^{\downarrow\uparrow}(\sigma,\mu,\phi)\ d\mu\ d\phi. \tag{1}$$

As our computations are performed on the longwave spectrum and for clear-sky conditions, we assume the atmosphere to be non-scattering, and we neglect the radiance of the sun. We also assume an isotropic thermal emission of gases. Under such assumptions, the radiance is independent of the azimuthal angle, thus we replace $I^{\downarrow\uparrow}(\sigma,\mu,\phi)$ by $I^{\downarrow\uparrow}(\sigma,\mu)$. By integration with

respect to $\phi$ we obtain

$$F^{\downarrow\uparrow}(\sigma) = 2\pi \int\limits_0^1 \mu I^{\downarrow\uparrow}(\sigma,\mu)\ d\mu. \tag{2}$$

---

[2]For a clarity purpose, the dependence on the wavenumber $\nu$ [unit: $cm^{-1}$] are not mentioned explicitly in variable names. For instance, where we should have written $F_\nu^{\downarrow}(\sigma_\nu)$, we simply write $F^{\downarrow}(\sigma)$. All variables referred to as monochromatic follow the same notation convention.





### 2.2.1 Downwelling flux

We first focus on the downwelling flux and the treatment of the upwelling flux will be described afterwards. Under the hypotheses of local thermodynamic equilibrium and local stratification of the atmosphere (plane-parallel geometry), the radiative transfer equation can be solved for the radiance. As we focus on the longwave only, the radiative contribution of space to the downwelling flux is negligible compared to the atmospheric contribution. Thus, the only remaining term in the solution of the radiative transfer equation is the atmospheric component:

$$I^{\downarrow}(\sigma,\mu) = \int_0^{\sigma} \frac{1}{\mu} B\left[T(\sigma')\right] \exp\left[-(\sigma-\sigma')/\mu\right] \, d\sigma'. \tag{3}$$

Thus, by injecting Eq. (3) into Eq. (2), we obtain

$$F^{\downarrow}(\sigma) = 2\pi \int_0^1 \int_0^{\sigma} B\left[T(\sigma')\right] \exp\left[-(\sigma-\sigma')/\mu\right] \, d\sigma' \, d\mu, \tag{4}$$

Where $B[T(\sigma')]$ is the Planck function [unit: $\mathrm{W.m^{-2}.sr^{-1}.(cm^{-1})^{-1}}$] evaluated at wavenumber $\nu$ (omitted in the notation) and at temperature $T(\sigma')$ [unit: K]. By switching the integration order, we can rewrite the downwelling flux as

$$F^{\downarrow}(\sigma) = 2\pi \int_0^{\sigma} B\left[T(\sigma')\right] E_2(\sigma-\sigma') \, d\sigma'. \tag{5}$$

Here, we have used the exponential integral of order $n$ defined for $n > 0$ and $x \geq 0$ as

$$E_n(x) = \int_0^1 \mu^{n-2} e^{-x/\mu} \, d\mu. \tag{6}$$

This formulation is commonly used to compute fluxes (Stamnes et al., 2017; Feigelson et al., 1991; Ridgway et al., 1991). The exponential integral functions can be evaluated numerically even though the computational time is significant in the context of a line-by-line flux calculation. To overcome this limitation, 4A-Flux uses a tabulation of the $E_n$ functions that is pre-calculated using the GNU Scientific Library (website: https://www.gnu.org/software/gsl/).

In order to compute the downwelling flux for the discretized model of the atmosphere used in 4A/OP model, we decompose the vertical integral into sums over every layer of the atmosphere. The atmospheric column is decomposed into $N+1$ levels, using subscripts denoted $k$ ranging from 0 to $N$, or $N$ layers, using subscripts denoted $k-\frac{1}{2}$ with $k$ ranging from 1 to $N$. We denote the value of the optical depth between the TOA and any level $k$ as $\sigma_k$. At the TOA $\sigma_0 = 0$ and at the surface $\sigma_N = \sigma_{BOA}$. At the TOA, the downwelling flux is supposed null, $F^{\downarrow}(\sigma_0) = 0$, and for every level $k > 0$, we have

$$F^{\downarrow}(\sigma_k) = 2\pi \sum_{i=1}^{k} \int_{\sigma_{i-1}}^{\sigma_i} B\left[T(\sigma')\right] E_2(\sigma_k - \sigma') \, d\sigma'. \tag{7}$$

To compute the atmospheric contributions of each layer to the fluxes (the integrand of Eq. (7)) a hypothesis has to be made about the variation of the Planck function at sublayer scale. The simplest hypothesis considers the Planck function as uniform





at the sublayer scale (isothermal layer) and evaluated at the average temperature of the layer $B\left[T\left(\sigma'\right)\right] = B[\bar{T}_{i-\frac{1}{2}}]$, where $\bar{T}_{i-\frac{1}{2}} = [T(\sigma_i) + T(\sigma_{i-1})]/2$. This hypothesis is a rather coarse approximation of the effective emission of the layer especially

when the optical depth is important. In such a case, the effective temperature is closer to the lower (resp. upper) boundary of the layer for the downwelling (resp. upwelling flux). A better hypothesis should take into account that the effective emission temperature is related to the optical depth of the layer. Thus, we have considered in our approach a sublayer variation of the Planck function as linear in optical depth (also referred to as $B(\sigma)$ *linear*). According to Ridgway et al. (1991), this approach dates back to Schuster (1905). Under such an assumption, the sublayer variation of the Planck function, in the layer $i - \frac{1}{2}$, and

for the downwelling flux at level $i$, is approximated by

$$B\left[T\left(\sigma'\right)\right] = B\left[T(\sigma_i)\right] + \left[\frac{2(B[\bar{T}_{i-\frac{1}{2}}] - B[T(\sigma_i)])}{\sigma_i - \sigma_{i-1}}\right](\sigma_i - \sigma'). \tag{8}$$

The hypothesis of an isothermal layer would lead to a systematic cold (resp. warm) bias in estimated downwelling (resp. upwelling) fluxes (Ridgway et al., 1991), whereas the $B(\sigma)$ *linear* assumption substantially reduces this bias (the remaining bias is discussed by Wiscombe (1976)). To better understand this approximation, one can verify that Eq. (8) is equivalent to

the isothermal layer hypothesis when the layer optical depth tends to zero and is equivalent to the Planck function taken at the lower (resp. upper) layer boundary temperature for the downwelling (resp. upwelling) flux when the layer optical depth tends to infinity. For a more detailed explanation the reader can refer to Clough et al. (1992). Thanks to the $B(\sigma)$ *linear* approximation, the downwelling flux can be integrated. To simplify the notations, we introduce the optical depth between two levels $i$ and $j$ as $\Delta_{i,j} = \sigma_i - \sigma_j$ (for $0 \le i, j \le N$). The downwelling flux, for every level $k > 0$, can finally be expressed as

$$
\begin{aligned}
F^{\downarrow}(\sigma_k) = &\ 2\pi \sum_{i=1}^{k} B\left[T\left(\sigma_i\right)\right] \left[E_3\left(\Delta_{k,i}\right) - E_3\left(\Delta_{k,i-1}\right)\right] \\
&+ 2\pi \sum_{i=1}^{k} 2(B[\bar{T}_{i-\frac{1}{2}}] - B[T(\sigma_i)]) \left[\frac{E_4\left(\Delta_{k,i}\right) - E_4\left(\Delta_{k,i-1}\right)}{\Delta_{i,i-1}} - E_3\left(\Delta_{k,i-1}\right)\right].
\end{aligned}
\tag{9}
$$


The first sum in Eq. (9) can be viewed as the cumulative downwelling flux emitted at the temperature of the lower boundary of every layer whereas the second sum brings the correction needed to account for the fact that the equivalent temperature of emission is somewhere between lower boundary temperature $T(\sigma_i)$ and the average layer temperature $\bar{T}_{i-\frac{1}{2}}$ depending on the value of the optical depth of the layer $\Delta_{i,i-1}$.

**2.2.2   Upwelling flux**

Now that the downwelling flux have been treated, we will focus on the monochromatic upwelling flux. Unlike for the downwelling flux of Eq. (4) where the downwelling radiance only accounts for the atmospheric component, the upwelling radiance must also account for the surface emission and the reflection of the downwelling flux on the surface. Thus, the upwelling flux





is written as

$$
F^{\uparrow}(\sigma) = 2\pi \overbrace{\int\limits_{0}^{1} \int\limits_{\sigma}^{\sigma_{BOA}} B[T(\sigma')] \exp\left[-(\sigma'-\sigma)/\mu\right] d\sigma' \, d\mu}^{\text{atmospheric component}}
$$

$$
+ 2\pi \overbrace{\int\limits_{0}^{1} \mu\epsilon B(T_s) \exp\left[-(\sigma_{BOA}-\sigma)/\mu\right] d\mu}^{\text{surface component}}
$$

$$
+ 2\pi \overbrace{\int\limits_{0}^{1} \mu\left\{(1-\epsilon)I^{\downarrow}(\sigma_{BOA},\mu)\exp\left[-(\sigma_{BOA}-\sigma)/\mu\right]\right\} d\mu.}^{\text{reflected component}}
\tag{10}
$$

The first term of this equation is the contribution of the atmospheric emission to the upwelling flux, which is analog to the downwelling flux of Eq. (4), replacing the boundaries of the integral and the argument of the exponential function. The second term is the contribution of the surface emission at temperature $T_s$ considering a surface spectral diffuse emittance $\epsilon$ [unit: dimensionless], at wavenumber $\nu$ (omitted in the notation). The third term represents the contribution of the downwelling flux that is reflected on the surface and consequently contributes to the upwelling flux. The presence of a reflected contribution to the upwelling flux is the consequence of the fact that the surface does not behave exactly as a black body. The flux emerging from the surface must take the spectral emittance of the surface into account. Under the assumption of specular reflection we have $I^{\uparrow}(\sigma_{BOA},\mu) = I^{\downarrow}(\sigma_{BOA},\mu)$.

For the atmospheric component at level $k$, denoted $F_{atmo}^{\uparrow}(\sigma_k)$, we follow the same rationale as we previously have for the downwelling flux. However, we have to rewrite Eq. (8), replacing $B[T(\sigma_i)]$ by $B[T(\sigma_{i-1})]$ and $(\sigma_i - \sigma')$ by $(\sigma' - \sigma_{i-1})$ to account for the change of direction of propagation. And, we also have to integrate the radiance from the BOA to level $\sigma_k$ instead of integrating from the TOA to level $\sigma_k$. We have $F_{atmo}^{\uparrow}(\sigma_N) = 0$ and, for every level $k < N$, we finally obtain

$$
F_{atmo}^{\uparrow}(\sigma_k) = 2\pi \sum_{i=k+1}^{N} B\left[T\left(\sigma_{i-1}\right)\right]\left[E_3\left(\Delta_{i-1,k}\right) - E_3\left(\Delta_{i,k}\right)\right]
$$
$$
+ 2\pi \sum_{i=k+1}^{N} 2(B[\bar{T}_{i-\frac{1}{2}}] - B[T(\sigma_{i-1})])\left[\frac{E_4\left(\Delta_{i-1,k}\right) - E_4\left(\Delta_{i,k}\right)}{\Delta_{i,i-1}} - E_3\left(\Delta_{i,k}\right)\right].
\tag{11}
$$

The surface component at level $k$, denoted $F_{surf}^{\uparrow}(\sigma_k)$, can be rewritten using the $E_3$ function as

$$
F_{surf}^{\uparrow}(\sigma_k) = 2\pi \, \epsilon \, B(T_s) E_3(\Delta_{N,k})
\tag{12}
$$





For the reflected component at level $k$, denoted $F^{\uparrow}_{refl}(\sigma_k)$, the integration of the third term of Eq. (10) leads to

$$F^{\uparrow}_{refl}(\sigma_k) = 2\pi(1-\epsilon) \sum_{i=1}^{N} B\left[T\left(\sigma_i\right)\right]\left[E_3\left(\Delta_{N,k}+\Delta_{N,i}\right) - E_3\left(\Delta_{N,k}+\Delta_{N,i-1}\right)\right]$$
$$+ 2\pi(1-\epsilon) \sum_{i=1}^{N} 2(B[\bar{T}_{i-\frac{1}{2}}] - B[T(\sigma_i)])\left[\frac{E_4\left(\Delta_{N,k}+\Delta_{N,i}\right) - E_4\left(\Delta_{N,k}+\Delta_{N,i-1}\right)}{\Delta_{i,i-1}} - E_3\left(\Delta_{N,k}+\Delta_{N,i-1}\right)\right].$$
(13)

This equation can be understood by comparison to the equation to the downwelling flux (Eq. (9)). The first difference is

the $(1-\epsilon)$ factors before the sums that correspond to the fraction of downwelling radiation that is reflected by the surface.

The second difference to Eq. (9) is that we sum all the layers from $1$ to $N$ because the reflected flux is a fraction of the

downwelling flux that reaches the BOA. Finally, the third difference is the argument of the $E_n$ functions. These arguments are

now incremented by $\Delta_{N,k}$ which corresponds to the optical depth from the surface to the level $k$ (upwelling propagation after

the reflection).

### 2.2.3 Radiative quantities

In 4A-Flux, the total upwelling flux is finally obtained by adding the three contributions described above:

$$F^{\uparrow}(\sigma_k) = F^{\uparrow}_{atmo}(\sigma_k) + F^{\uparrow}_{surf}(\sigma_k) + F^{\uparrow}_{refl}(\sigma_k)$$
(14)

Once the monochromatic hemispherical fluxes are computed using Eq. (9) and Eq. (14), 4A-Flux computes the net flux

denoted $F(\sigma_k)$ using the following convention

$$F(\sigma_k) = F^{\uparrow}(\sigma_k) - F^{\downarrow}(\sigma_k).$$
(15)

The monochromatic OLR is simply defined as the net flux at the TOA $F(\sigma_0)$.

The monochromatic vertical heating rate, denoted $H$ (omitted spectral dependence), is defined as the divergence of the net

flux and is conventionally expressed in units of Kelvin per day [unit: $\mathrm{K.day}^{-1}$]. In the plane-parallel geometry, it is defined as

$$H = \frac{g}{c_p}\frac{\partial F}{\partial p}$$
(16)

The heating rate is usually negative for the longwave and thus the expression *cooling rate* is sometimes preferred to the

expression *heating rate* that we are using here. This quantity represents the rate of change of temperature caused by radiative

properties of the atmosphere. In 4A-Flux, the computation of the heating rate at every layer $k-\frac{1}{2}$ is performed from the net

flux of adjacent levels $k$ and $k-1$ by calculating the difference quotient

$$H(\sigma_{k-\frac{1}{2}}) = \frac{g}{c_p}\frac{F(\sigma_{k-1}) - F(\sigma_k)}{p(\sigma_{k-1}) - p(\sigma_k)}.$$
(17)

We then convert the heating rate from SI units to commonly used $\mathrm{K.day}^{-1}$ unit.



**Table 1.** CPU time of the complete radiative transfer including calculation of radiative quantities, for a single atmospheric situation.

| Atmospheric situation | CPU time (vertical profiles) | CPU time (OLR only) |
|---|---|---|
| 43 vertical levels (TIGR situation) | 26 min 57 sec | 1 min 54 sec |
| 61 vertical levels (RFMIP situation) | 1 h 0 min 34 sec | 2 min 24 sec |

Computations are performed sequentially on a quad-pro AMD Opteron 6378 16-core, 2.4 GHz and 256 Gb of RAM.
TIGR situations are described in sect. 4.1 and RFMIP situations are described in sect. 3.

195  The computation time of the total radiative transfer including the calculation of the hemispherical fluxes, the net flux and the heating rate profile is indicated in Table 1. The longwave fluxes are computed at the spectral resolution of 4A/OP model $(5 \cdot 10^{-4} \text{ cm}^{-1})$ from 10 to 3250 cm$^{-1}$. The computation in the "OLR only" option is a lot faster than the complete vertical profile computation as the contributions of each atmospheric layers are only computed for the TOA level. The dependence to the number of levels of the atmospheric model is approximately linear for in the "OLR only" option and approximately 200 quadratic for the full vertical profile computation (omitting the initialization times).

## 3 Evaluation of 4A-Flux over the RFMIP database

The Radiative Forcing Model Intercomparison Project (RFMIP[3] ; protocol paper: Pincus et al. (2016)) is one of the 23 inter-comparison projects supported by the World Climate Research Program (WRCP) in the context of the 6th phase of the Coupled Model Intercomparison Project (CMIP6, see Eyring et al. (2016)). The objective of the experiment RFMIP-IRF (RFMIP In-205 stantaneous Radiative Forcing from greenhouse gases) is to characterize the accuracy of the parameterization of the IRF used in climate models under clear-sky and aerosol-free conditions. This characterization is performed in comparison to line-by-line radiative transfer models that are recognized for their very high accuracy. Thanks to the new implementation of 4A-Flux module, 4A/OP radiative transfer model contributed to RFMIP-IRF by providing longwave downwelling and upwelling fluxes. Calculations of radiative forcing by greenhouse gases with the six benchmark models that are participating to RFMIP-IRF, 210 4A/OP included, are listed in Table 2. These models and the radiative forcing evaluation results are presented in Pincus et al. (2020).

  RFMIP provides a sample of 100 atmospheric situations from reanalysis of present-day conditions containing profiles of pressure, temperature, greenhouse gas concentration and surface properties. When averaged using the provided weights, the fluxes estimated from individual profiles can be used to estimate the time-averaged global-mean fluxes with very small sampling 215 errors (relative to disagreements between models). Along with the 100 present-day atmospheric situations, 17 perturbations around the present-day are also provided leading to a total sample size of 1800 atmospheric situations (perturbations are listed in appendix A). All six benchmark models have provided longwave integrated upwelling and downwelling flux profiles for the whole set of atmospheric profiles.

---

[3]RFMIP website: https://rfmip.leeds.ac.uk/





**Table 2.** List of the six benchmark models that have contributed to RFMIP-IRF and that are compared in this study.

| Model Name | Institution | Type of model | References |
|---|---|---|---|
| 4A/OP v1.5 / 4A-Flux v1.0 | IPSL | line-by-line | Cheruy et al. (1995), Scott and Chedin (1981) / this work |
| ARTS-2-3 | UHH | line-by-line | Buehler et al. (2018) |
| GFDL-GRTCODE | NOAA-GFDL | line-by-line | |
| GFDL-RFM-DISORT | NOAA-GFDL | line-by-line | Dudhia (2017) |
| HadGEM3-GC31-LL | MOHC NERC | narrow-band | Edwards and Slingo (1996), Walters et al. (2019) |
| LBLRTM-12-8 | AER | line-by-line | Clough et al. (2005) |

IPSL: Institut Pierre Simon Laplace ; UHH: Universitat Hamburg ; NOAA-GFDL: National Oceanic and Atmospheric Administration, Geophysical Fluid Dynamics Laboratory ; MOHC: Met Office Hadley Centre ; NERC: Natural Environment Research Council ; AER: Research and Climate Group, Atmospheric and Environmental Research

Participating models have usually provided several sets of spectrally integrated longwave flux profiles, for slightly different
sets of greenhouse gases, called forcing variants as described in Meinshausen et al. (2017). For the first forcing variant (denoted
f1), models are free to use any subset of the 43 greenhouse gases specified in Meinshausen et al. (2017). For forcing variant
f1, 4A/OP take into account 16 gases out of the 43 gases included into the RFMIP database[4]. Both second and third forcing
variant (respectively denoted f2 and f3) use $CO_2$, $CH_4$ and $N_2O$. In addition to those three gases, forcing variant f2 also uses
CFC-12 concentration and a modified concentration of CFC-11 to account for all omitted gases whereas forcing variant f3 uses
modified concentrations of CFC-11 and HFC-134a to represent all omitted gases.

Some participating models have also provided several sets of results based on slightly different model configurations, called
physics variants. As described in Pincus et al. (2020), the model ARTS 2.3 only includes $CO_2$ line-mixing for physics variant
p2 and not for physics variant p1. The model HadGEM3-GC3.1 physics variant p1 uses a lower spectral resolution than physics
variant p2 that corresponds to the high resolution configuration. The physics variant p3 is the high resolution configuration with
MT_CKD 3.2 treatment of the water vapor continuum instead of CAVIAR continuum.

Using RFMIP database, we have evaluated the downwelling and upwelling fluxes computed with 4A-Flux module of 4A/OP
model against calculations of other participating radiative transfer models. To perform this comparison, we have averaged
downwelling and upwelling fluxes for the 1800 atmospheric situations, using the provided weights to average the 100 atmo-
spheric situations and uniformly averaging the 18 perturbations. For any given couple of model and physics variant, we have
also averaged the downwelling and upwelling fluxes for every provided forcing variants. Forcing variants are averaged for
a clarity purpose as for every model, variability between forcing variants is smaller than the variability between models and
physics variants (not shown here). For HadGEM3-GC3.1, we have not retained the physics variant p1 for the comparison as it
uses a lower spectral resolution and thus would not usefully serve as a reference. Figure 2 shows the distance $\Delta F_i^{\uparrow\downarrow}(p)$ from the
mean downwelling and upwelling flux profiles of each model $i$ to the mean of all profiles for the eight retained model/physics

---

[4]The gases that have been taken into account in 4A/OP simulation for forcing variant f1 are $H_2O$, $O_3$, $CO_2$, $CH_4$, $N_2O$, CO, $O_2$, $N_2$, $CH_3Cl$, $SF_6$,
CFC-11, CFC-12, $CCl_4$, CFC-113, HCFC-22 and HFC-134a.

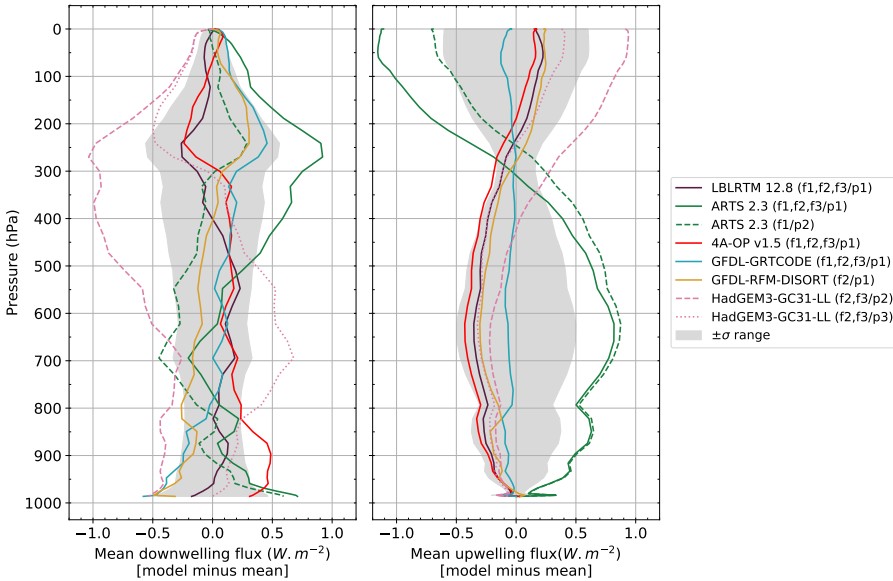

**Figure 2.** Comparison of net flux profiles between line-by-line models participating to RFMIP-IRF. The left (resp. right) panel shows the distance to the mean of the mean downwelling (resp. upwelling) flux profiles for every couple of model and physics variant. The averaging is first performed on the 100 atmospheric profiles using provided weights and then on the 18 perturbation (uniformly weighted). Finally, for every couple of model and physics variant, all available forcing variants are averaged. The $\pm 1$ standard deviation range ($\pm \sigma$) is represented by the gray shaded area.

variant couples. This distance is computed as

$$\Delta F_i^{\uparrow \downarrow}(p) = F_i^{\uparrow \downarrow}(p) - \frac{1}{8} \sum_i F_i^{\uparrow \downarrow}(p). \tag{18}$$

The $\pm 1$ standard deviation range between models is also displayed in Figure 2 (gray shaded area). We first notice the strong agreement between all compared models. The standard deviation between models is smaller than $0.61$ W.m$^{-2}$ for the downwelling flux and $0.75$ W.m$^{-2}$ for the upwelling flux. 4A/OP model outputs are included in the $\pm 1$ standard deviation range

almost at every level with the exceptions of the $993 - 976$ hPa pressure range for the downwelling flux and of the $300 - 401$ hPa pressure range for the upwelling flux where they nonetheless stay close to the $\pm 1$ standard deviation range. The extremum difference for 4A/OP model ranges from $-0.24$ to $0.49$ W.m$^{-2}$ for the downwelling flux profile and from $-0.43$ to $0.20$ W.m$^{-2}$ for the upwelling flux profile.

    The OLR estimations are compared using the same averaging methodology. However, we have not averaged forcing variants

to give more detailed results. The statistics for all models are calculated by first averaging forcing variants for every [model, physics variant] couples, and then we average these couples together. Table 3 presents these statistics (model minus 4A/OP). The mean difference between 4A/OP and other models is $-0.148$ W.m$^{-2}$ and the mean standard deviation is $0.218$ W.m$^{-2}$. This shows a very good agreement between 4A/OP and other models. Note that 4A/OP simulates OLR that are very close





**Table 3.** Means and standard deviations of OLR differences (model minus 4A/OP) in $\mathrm{W.m^{-2}}$.

| Model Name | Physics variant | Forcing variant | Difference mean | Difference standard deviation |
|---|---|---|---|---|
| LBLRTM 12.8 | p1 | f1 | −0.019 | 0.206 |
| LBLRTM 12.8 | p1 | f2 | 0.047 | 0.083 |
| LBLRTM 12.8 | p1 | f3 | 0.007 | 0.098 |
| ARTS 2.3 | p1 | f1 | −1.328 | 0.478 |
| ARTS 2.3 | p1 | f2 | −1.214 | 0.423 |
| ARTS 2.3 | p1 | f3 | −1.242 | 0.432 |
| ARTS 2.3 | p2 | f1 | −0.888 | 0.412 |
| GFDL-GRTCODE | p1 | f1 | −0.246 | 0.243 |
| GFDL-GRTCODE | p1 | f2 | −0.167 | 0.176 |
| GFDL-GRTCODE | p1 | f3 | −0.196 | 0.179 |
| GFDL-RFM-DISORT | p1 | f2 | 0.118 | 0.167 |
| HadGEM3-GC31-LL | p2 | f2 | 0.799 | 0.266 |
| HadGEM3-GC31-LL | p2 | f3 | 0.770 | 0.265 |
| HadGEM3-GC31-LL | p3 | f2 | 0.266 | 0.217 |
| HadGEM3-GC31-LL | p3 | f3 | 0.237 | 0.218 |
| Mean all models | | | −0.148 | 0.218 |

to the LBLRTM simulations especially for forcing index f2 and f3 where the exact same gas list has been simulated. In the

forcing index f3, the mean difference between 4A/OP and LBLRTM is $0.007\ \mathrm{W.m^{-2}}$ with a difference standard deviation of no more than $0.098\ \mathrm{W.m^{-2}}$. The difference to all models is always smaller than $1\ \mathrm{W.m^{-2}}$ except for ARTS physics index p1 that does not include $CO_2$ line-mixing. The greater observed difference in standard deviation noticeable for forcing variant f1 compared to f2 and f3 variants is due to the gases that are not yet implemented in 4A/OP and thus not simulated in f1 whereas all gases of f2 and f3 are identical between models. This is further explained in appendix A where experiments, forcing and

physics variants are compared without any averaging.

The results of this comparison show a very good agreement of 4A/OP with the five other participating models. The 4A/OP simulation yields very close results to LBLRTM especially for the longwave upwelling flux and the OLR. This good agreement while the two models differ completely both in numerical methods and in the spectroscopic databases used (GEISA for 4A/OP, HITRAN for LBLRTM) are important factors in the confidence that we can have in their results. Furthermore, these deviations

between models are well below the sensitivities to typical perturbations of surface and atmospheric parameters that we will describe next.





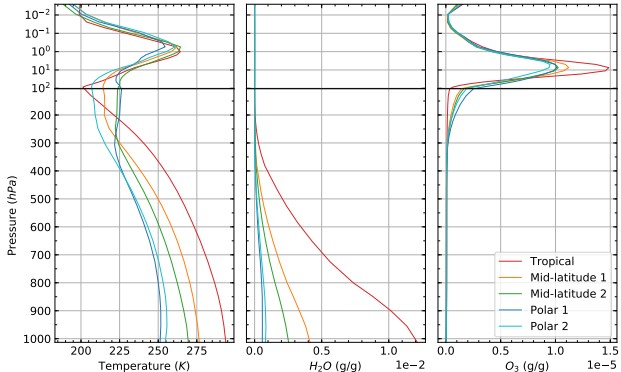

**Figure 3.** Average atmospheric profiles of temperature (left), water vapor concentration (center) and ozone concentration (right) of the mean atmosphere of each of the five classes of TIGR. The concentrations are given in grams of gas per gram of dry air. The upper panels present the profiles on a logarithmic pressure scale and the lower panels present them on a linear pressure scale.

## 4 Sensitivity of Radiative Fluxes and Vertical Heating Rate to Temperature and Composition

In this section, we present a first application of the 4A-Flux module newly implemented in the radiative transfer code 4A/OP. Following Clough and Iacono (1995), we perform a sensitivity study of the OLR and the vertical heating rate to various atmo-
spheric and surface parameters. This sensitivity is based on the Thermodynamic Initial Guess Retrieval (TIGR) atmospheric database that is developed at the *Laboratoire de Météorologie Dynamique*. We will first present the TIGR database and then describe the sensitivity study that we have performed on the OLR and on the longwave vertical heating rate.

### 4.1 TIGR atmospheric database

The Thermodynamic Initial Guess Retrieval (TIGR) atmospheric database is a climatological library containing 2311 atmo-
spheric situations that have been carefully selected by statistical methods from radiosonde reports (Chédin et al., 1985; Achard, 1991; Chevallier et al., 1998). The atmospheric samples are described by their vertical profiles of temperature, water vapor and ozone concentration on a pressure grid of 43 levels. The pressure levels range from the surface at $1013\,\mathrm{hPa}$ up to $0.0026\,\mathrm{hPa}$ and the density of points along the vertical increases while pressure decreases to correctly account for the upper atmospheric layers. The 2311 atmospheres are classified into five airmass types. The *Tropical* class contains situations in the tropics, the
*Midlat 1* class contains mid-latitude situations, the *Midlat 2* class contains cold mid-latitude and summer polar situations, the *Polar 1* class contains Northern Hemisphere very cold polar situations, the *Polar 2* class contains both hemispheres winter polar situations. This classification has been performed by a hierarchical ascending classification depending on their virtual temperature profiles (Achard, 1991; Chédin et al., 1994).

Figure 3 represents the mean profiles of temperature, water vapor and ozone concentration computed on each of the five
classes of TIGR database. OLR calculated by 4A-Flux using these five averaged atmospheres are represented in Figure 4. Except for temperature, water vapor and ozone, other simulated variable are identical for all five atmospheres. Table 4 presents



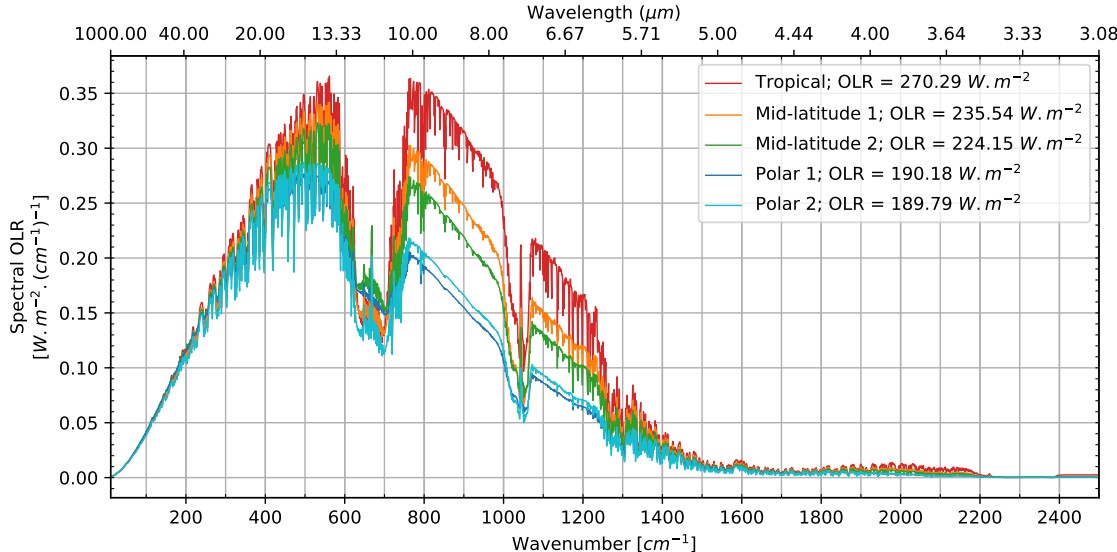

**Figure 4.** Spectral OLR at a resolution of $1 \ \mathrm{cm}^{-1}$ computed for the mean atmosphere of each of the five classes of TIGR. Spectrally integrated OLR on the $10 - 2500 \ \mathrm{cm}^{-1}$ range are displayed in the legend.

the concentration of Well-Mixed GreenHouse Gases (WMGHGs) considered as a reference for our simulations. For all the simulations we perform here, the surface spectral diffuse emittance is set at a constant value of $0.98$ on the entire longwave spectrum. For more concision, the averaged atmospheres of the five classes will be called by the name of their class (e.g.

*Tropical atmosphere* or *Tropical situation* instead of *mean atmosphere of the Tropical class*).

The Tropical temperature at the surface and in the troposphere is significantly higher than all other classes, especially both Polar situations that exhibits lower temperatures. Higher temperatures lead to a higher OLR especially noticeable in the atmospheric IR window. This difference is lower where the absorption is important such as on the carbon dioxide $\nu_2$ absorption band. At the tropopause, the temperature profile of the Tropical situation is the lowest. Above the tropopause, the temperature

profiles are closer together.

Mainly present in the troposphere, water vapor is 15 times more concentrated at the surface of the Tropical atmosphere than at the surface of both Polar atmospheres. The Tropical atmosphere has the highest water vapor content compared to all other classes especially compared to the dry atmospheres of the Polar classes. Consequently, we observe less impact of the water vapor absorption on the OLR spectra of the drier atmospheres.

In all atmospheres, ozone concentration is high the stratosphere and in the mesosphere and relatively low in the troposphere. However, it is less spread and peaks higher in the Tropical class compared to other classes. Thus, the OLR of the mean tropical atmosphere is more impacted on the ozone absorption band than all other classes.

As they represent very different and opposite situations, we will only focus next on two specific classes: the hot and humid Tropical class and the cold and dry Polar 2 class. We will now present the results of a sensitivity study of the OLR and





**Table 4.** Concentration of WMGHG and minor gases used for the simulation. The concentrations of all these gases are assumed vertically constant.

| Gas | Concentration [g.g$^{-1}$ of dry air] | Concentration [ppmv] |
|---|---|---|
| $CO_2$ | 6.00e-04 | 3.95e+02 |
| $CH_4$ | 1.03e-06 | 1.86e+00 |
| $N_2O$ | 4.92e-07 | 3.24e-01 |
| CO | 9.70e-08 | 1.00e-01 |
| $O_2$ | 2.31e-01 | 2.09e+05 |
| $N_2$ | 7.55e-01 | 7.81e+05 |
| $CH_3Cl$ | 1.21e-09 | 7.01e-04 |
| $SF_6$ | 2.81e-11 | 5.57e-06 |
| CFC11 | 1.28e-09 | 2.69e-04 |
| CFC12 | 2.11e-09 | 5.05e-04 |
| $CCl_4$ | 7.02e-10 | 1.32e-04 |
| CFC113 | 4.02e-10 | 6.21e-05 |
| HCFC22 | 6.85e-10 | 2.30e-04 |
| HFC134a | 1.48e-09 | 4.21e-04 |

longwave heating rate profile to several perturbations of surface and atmospheric parameters using TIGR Tropical and Polar 2 atmospheric situations as input of 4A/OP model.

## 4.2 OLR sensitivity

Here we seek to quantify the effects of perturbations applied to atmospheric and surface parameters on the OLR spectrum and on the spectrally integrated OLR. To perform this analysis, we have calculated the OLR differences between a reference

atmosphere and a modified atmosphere affected by a series of perturbations. The reference atmospheres used here are computed with the TIGR mean Tropical and mean Polar 2 atmospheric situations that are presented in sect. 4.1. We first present the reference OLR spectra in both Tropical and Polar 2 situations, and then we present the effects of several thermodynamic and composition perturbations. We will describe the effects of an increase of surface temperature of 1 K above the reference and of a vertically uniform increase of the atmospheric temperature of 1 K. And, we will analyze the effects a doubling of

the concentrations of the three main WMGHG : carbon dioxide, methane and nitrous oxide. Each perturbation is studied independently of one another and are applied uniformly on the vertical profile. And then, we will examine the effects of a $+1\%$ vertically uniform increase of water vapor concentration and of ozone concentration.

Figure 5 presents the OLR spectra at resolution 1 cm$^{-1}$ simulated by 4A-Flux module in the Tropical situation on the panel (a), and in the Polar 2 situation on the panel (b). Figure 5 also shows the effects of studied perturbations on OLR as a function of



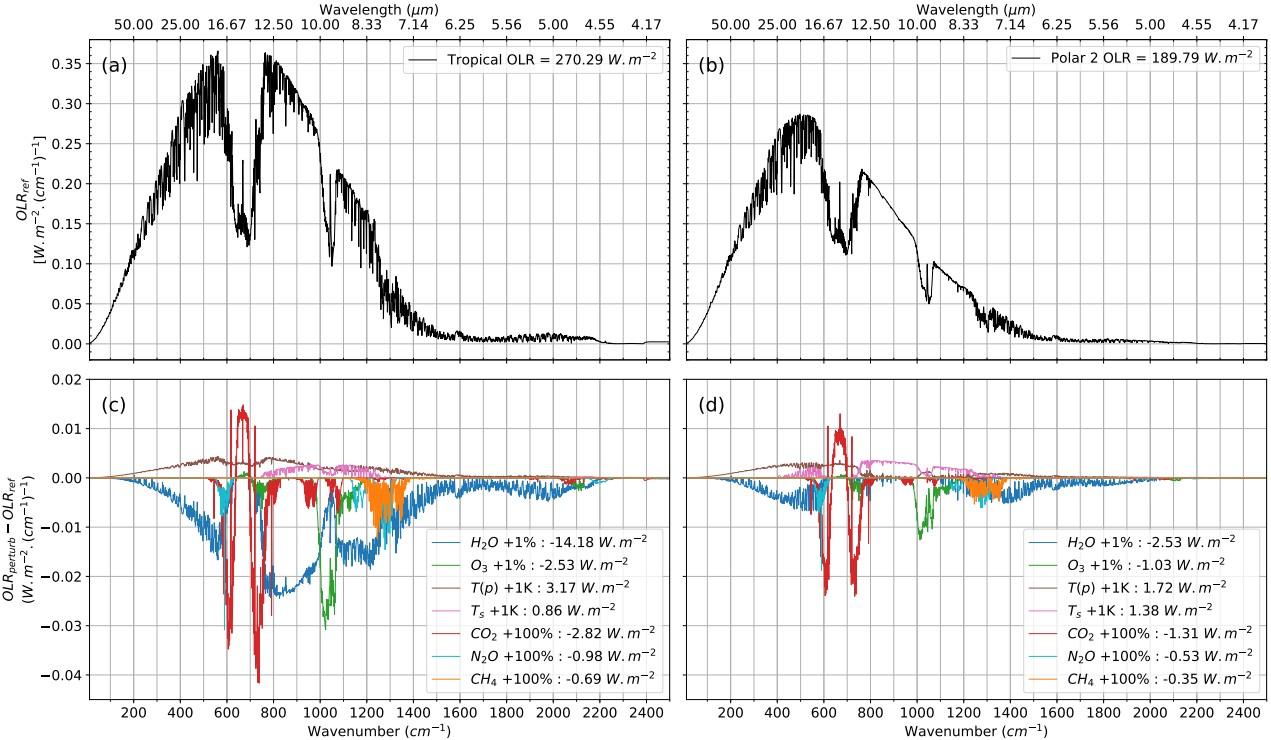

**Figure 5.** Spectral OLR at a spectral resolution of $1 \text{ cm}^{-1}$ computed for the TIGR mean Tropical atmosphere (a) and TIGR mean Polar 2 atmosphere (b). Corresponding OLR values spectrally integrated between $10$ and $2500 \text{ cm}^{-1}$ are provided on the legends. Spectral OLR sensitivity to temperature and composition change for the TIGR mean Tropical (c) and mean Polar 2 (d) atmospheres. Results are expressed as $[OLR_{\text{perturb.}} - OLR_{\text{ref.}}]$ as a function of the wavenumber. The spectrally integrated sensitivities are specified on the legends and in Table 5.

the wavenumber in the Tropical situation on the panel (c), and in the Polar 2 situation on the panel (d) and Table 5 summarizes the spectrally integrated OLR variation caused by these perturbations.

The Tropical OLR spectrum and longwave integrated OLR (panel (a)) are greater than the Polar 2 OLR spectrum (panel (b)). This is mainly due to the substantially higher surface and atmospheric temperatures of the Tropical situation compared to the Polar 2 situation (almost $40$ K difference between the surface temperatures). The well-known signatures of the main longwave

active gases are also clearly visible on the Figure 5: Carbon dioxide ($580 - 750 \text{ cm}^{-1}$, $2200 - 2400 \text{ cm}^{-1}$), ozone ($980 - 1080$ $\text{cm}^{-1}$), water vapor (especially far infrared and large band around $1600 \text{ cm}^{-1}$), methane and nitrous oxide ($1250 - 1350$ $\text{cm}^{-1}$).

According to Eq. 12, the increase of the surface temperature is expected to lead to an increase of the surface Planck emission for all wavenumber at the BOA. At the TOA, the OLR is expected to increase only in the atmospheric IR window. This is

well verified here on panels (c) and (d) (pink curve). The Tropical atmosphere being optically thicker (higher humidity), the OLR sensitivity is spectrally confined to the infrared atmospheric window compared to the Polar 2 atmosphere. Because the





**Table 5.** Spectrally integrated OLR sensitivity to the variations of thermodynamic and composition variables.

| Variable | Perturbation | OLR Sensitivity Tropical | OLR Sensitivity Polar 2 |
|---|---|---|---|
| Temperature profile | $+1$ K | $3.17$ W.m$^{-2}$ | $1.72$ W.m$^{-2}$ |
| Surface temperature | $+1$ K | $0.86$ W.m$^{-2}$ | $1.38$ W.m$^{-2}$ |
| Water vapor concentration | $+1$ % | $-14.18$ W.m$^{-2}$ | $-2.53$ W.m$^{-2}$ |
| Ozone concentration | $+1$ % | $-2.53$ W.m$^{-2}$ | $-1.03$ W.m$^{-2}$ |
| Carbon dioxide concentration | $+100$ % (790 ppmv) | $-2.82$ W.m$^{-2}$ | $-0.31$ W.m$^{-2}$ |
| Nitrous oxide concentration | $+100$ % (648 ppbv) | $-0.98$ W.m$^{-2}$ | $-0.53$ W.m$^{-2}$ |
| Methane concentration | $+100$ % (3720 ppbv) | $-0.69$ W.m$^{-2}$ | $-0.35$ W.m$^{-2}$ |

atmospheric window spectral range is relatively free from absorbers other than water vapor, a lower concentration of water vapor leads to a higher OLR sensitivity to the surface temperature in this spectral range.

The vertically uniform increase of the atmospheric temperature profile is expected to lead to an increase of the Planck emission of every layer of the atmosphere (Eq. 11), leading to an increase of the OLR on bands where there is gas absorption (assuming the absorption lines of the gases are marginally impacted by the temperature perturbation). This is also verified in Figure 5. In the Tropical case, the increase extends on the whole longwave spectrum. However, for the dry Polar 2 atmosphere, there is very low absorption/emission on the infrared window.

The increase of the WMGHG concentration leads to a decrease of the surface contribution to the OLR (Eq. 12) as the total
surface-to-space optical depth increases. For the contribution of the atmospheric layers to the OLR however (Eq. 11), there is a competition between the increase of the thermal emission of atmospheric layers (increasing the OLR) and the increase of absorption by gases (decreasing the OLR). The $\nu_2$ absorption band of $CO_2$ shows a remarkable pattern. The increase of $CO_2$ causes an OLR decrease in the band wings, where the absorption dominates, however it causes an OLR increase in the band center, where the emission dominates. Integrated over the entire longwave, the absorption dominates causing an OLR decrease
on the longwave. The sensitivity of $N_2O$ ($550 - 640$ cm$^{-1}$, $1140 - 1320$ cm$^{-1}$) is dominated by the absorption, and we don't observe any local increase of OLR. The $CH_4$ sensitivity ($1200 - 1370$ cm$^{-1}$) is also dominated by the absorption.

Unlike for the WMGHG, water vapor absorbs on the entire longwave region. The sensitivity to water vapor concentration is especially important in the Tropical situation, and less important in the drier Polar 2 situation. In the Tropical case, the OLR decreases almost everywhere on the longwave spectrum, except where the $\nu_2$ carbon dioxide band dominates the absorption.
In the Polar 2 case, the sensitivity to water vapor almost completely disappear in the atmospheric window. A reduction in water vapor sensitivity is also notable on spectral bands where other gases absorb and compete water vapor absorption ($O_3$ on $1000 - 1080$ cm$^{-1}$ and both $CH_4$ and $N_2O$ on $1250 - 1350$ cm$^{-1}$). Water vapor being more concentrated in the lower troposphere, the WMGHG play a relatively greater part at the TOA.





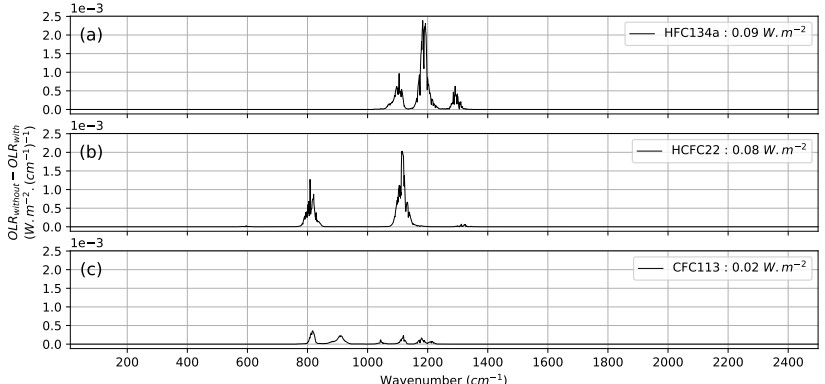

**Figure 6.** Spectral OLR variation due to a complete removal of atmospheric HFC134a (a), HCFC22 (b) and CFC113 (c), calculated on TIGR mean Tropical atmosphere. The total longwave OLR variations are reported on the legends. Note that the y-axis ranges from 0 to $2.5 \cdot 10^{-3}$ W.m$^{-2}$.(cm$^{-1}$)$^{-1}$

An increase in ozone concentration leads to an OLR reduction ($600 - 800$ cm$^{-1}$, $980 - 1080$ cm$^{-1}$). Ozone being more
concentrated in the stratosphere around 100 hPa, an increase of its concentration leads to a decrease of the surface and the
tropospheric and lower stratospheric contributions to the OLR. The increase of the emission to space is also present but remain
small as it can be seen in the $600 - 700$ cm$^{-1}$ range where there is a local increase of the spectral OLR.

Figure 6 shows the OLR sensitivity to three anthropogenic trace gases : HFC134a, HCFC22 and CFC113. Even if their
individual contributions to the total OLR are small, these three gases together represent $0.19$ W.m$^{-2}$ for the tropical atmosphere
which is approximately the standard deviation between 4A/OP and other RFMIP benchmark models. However, as demonstrated
and discussed by Pincus et al. (2020), the standard deviation between the six RFMIP benchmark models in terms of radiative
forcing is less than $0.025$ W.m$^{-2}$ — way below the standard deviation obtained on fluxes because of the cancellations when
computing the radiative forcing as a difference of hemispheric fluxes (Pincus et al., 2020; Mlynczak et al., 2016). Accounting
for more minor gases in line-by-line radiative transfer models could also improve our understanding of model intercomparison
discrepancies.

These applications show that 4A-Flux module of 4A/OP can be used to study OLR sensitivity to thermodynamic variables
and composition of the atmosphere. Thanks to its very fine spectral resolution (down to $5 \cdot 10^{-4}$ cm$^{-1}$), 4A/OP can help
evaluate the subtle impacts of minor longwave active gases such as HFC134a, HCFC22 and CFC113.

## 4.3 Vertical heating rate sensitivity

In this section, we focus on the sensitivity of the vertical heating rate profile caused by a vertically uniform increase of
1% of water vapor concentration and caused by a doubling in carbon dioxide concentration using the TIGR mean Tropical
atmospheric situation and the TIGR Polar 2 atmospheric situation as reference atmospheres. First, we describe the longwave





heating rate vertical and spectral distribution for the two reference atmospheric situations, and then we analyze the sensitivities that are calculated as the difference between the perturbed atmospheric state and the reference state: $H_{\mathrm{perturb.}}(p) - H_{\mathrm{ref.}}(p)$.

Figure 7 present the vertical heating rate as computed by 4A-Flux for the Tropical (panels (a1) to (c1)) and Polar 2 (panels (d1) to (f1)) atmospheric situations. The panels (a1) and (d1) show the vertical heating rate spectrum as a function of the wavenumber (horizontal axis) and the pressure level (vertical axis). The panels (b1) and (e1) show the heating rate profile spectrally integrated between 10 and $2500 \, \mathrm{cm}^{-1}$. On the panel (c1) and (f1), the range of the x-axis has been modified to focus on the tropospheric variations. On all panels, the vertical representation has been divided into two separate pressure scales. The
lower sub-panels, where the pressure scale is linear, highlight the tropospheric heating rate and the upper sub-panels, where the pressure scale is logarithmic, give a better insight into the heating rate above the tropopause. The color scale has been chosen to emphasize the main radiative signature of the different gases at multiple scales. Cool colors represent a net cooling while warm colors represent a net heating, except magenta which represents extreme cooling values.

Tropical and Polar 2 longwave integrated heating rate profiles show similarities (panels (b1), (c1), (e1) and (f1) in Figure
7). The heating rate profiles are negative almost everywhere on the vertical and show a characteristic shape. In both cases, we notice that the profiles are almost constant in the troposphere before increasing around a certain pressure level, referred to as the *kink*, that is recognized to be associated to several atmospheric processes (Jeevanjee and Fueglistaler, 2020; Hartmann and Larson, 2002). Above the kink, the heating rate increases and reaches its maximum value around $100 \, \mathrm{hPa}$ where the heating rate is close to $0 \, \mathrm{K.day}^{-1}$. Above $100 \, \mathrm{hPa}$, the heating rate profile decreases again until $1 \, \mathrm{hPa}$, then increases until $0.1 \, \mathrm{hPa}$
to finally decrease at the TOA. We also notice differences of heating rate profiles between the two atmospheric situation. First, the values of the heating rate until the kink are not the same (around $-2 \, \mathrm{K.day}^{-1}$ in the Tropical case, and $-1 \, \mathrm{K.day}^{-1}$ in the Polar 2 case). Second, the kink occurs around $250 \, \mathrm{hPa}$ in the Tropical atmosphere and around the $400 \, \mathrm{hPa}$ in the Polar 2 atmosphere, which approximately corroborates the $T_{kink} = 220 \, \mathrm{K}$ specified in Jeevanjee and Fueglistaler (2020). Above the kink, we notice a greater slope in the Tropical atmosphere than in the Polar 2 atmosphere.

The characteristic variations of the heating rate profile previously described can be further analyzed using the spectral and vertical distribution of absorption by longwave active gases. The representation on panel (a1) and (d1) of Figure 7 give insight into these distributions. In both atmospheric situations, the spectra of heating rate profile display similarities. We observe the signatures of the greenhouse gases accounted in the model. The water vapor effects dominates the Earth longwave cooling as it absorbs on a wide portion of the longwave spectrum. This is primarily due to the strong absorption of the pure rotational
band peaking in the $100 - 200 \, \mathrm{cm}^{-1}$ range but extending from $10 - 1000 \, \mathrm{cm}^{-1}$ range in the Tropical case. The water vapor effects are also caused by the relatively lower absorption by the rotational-vibrational band centered on $1600 \, \mathrm{cm}^{-1}$, and by the continuum of absorption. The radiative cooling induced by the $\nu_2$ absorption band of carbon dioxide is dominant in the $580 - 750 \, \mathrm{cm}^{-1}$ range. Due to the very high absorption of carbon dioxide in the band center, most of the radiative cooling occurs in the stratosphere. The cooling is very small under the tropopause temperature inversion, where a net heating occurs.
Above this tropopause heating, an intense cooling occurs in the stratosphere and mesosphere until the TOA. On the band wings of the carbon dioxide $\nu_2$ absorption band, we observe a net cooling on the whole profile except at the tropopause in the Tropical case where we observe a net heating. We observe the effects of the ozone absorption in its main absorption region,



**Figure 7.** Top row: Vertical heating rate computed by 4A-Flux using TIGR mean Tropical (resp. Polar 2) atmospheric situation on the left (resp. right) panels. Panels (a1) and (d1) present the spectral heating rate profile as a function of the wavenumber and the pressure level. Heating rate profile integrated on the $10 - 2500 \ \mathrm{cm}^{-1}$ spectral range is represented on panels (b1) and (e1) and a focus on the tropospheric variations is displayed on panel (c1) and (f1). Middle row: Vertical heating rate sensitivity to a 1% increase of $H_2O$ concentration. Panels (a2) and (d2) present the spectral heating rate sensitivity profiles $[H_{\mathrm{perturb.}}(p) - H_{\mathrm{ref.}}(p)]$ as a function of the wavenumber (horizontal axis) and the pressure level (vertical axis). Heating rate sensitivity profile integrated on the $10 - 2500 \ \mathrm{cm}^{-1}$ spectral range are represented on panel (b2) and (e2) and a focus on the tropospheric variations is displayed on panels (c2) and (f2). Bottom row : analog to middle row for a doubling of $CO_2$ concentration (instead of $H_2O$).



in the $980 - 1080 \ \mathrm{cm}^{-1}$ range. A strong heating occurs in the upper troposphere, tropopause, lower stratosphere and high in the mesosphere and an intense cooling in the upper stratosphere and stratopause. The same pattern, with a lower amplitude,

appears in the $1250 - 1350 \ \mathrm{cm}^{-1}$ range where methane and nitrous oxide absorb. The main differences of heating rate profile spectrum between the Tropical and the Polar 2 atmospheres is the amplitude of cooling and the pressure range at which the cooling occurs. In the troposphere of the Tropical atmosphere, the cooling is more intense due to the association of a steeper temperature gradient and a higher water vapor concentration. We also notice that, due to the more vertically spread ozone concentration profile in the Polar 2 atmosphere compared to the Tropical atmosphere, the ozone cooling pressure range is

wider.

The vertical heating rate sensitivity to a 1% increase of water vapor concentration are presented in Figure 7 with panels (a2), (b2) and (c2) for the Tropical atmosphere, and panels (d2), (e2) and (f2) for the Polar 2 atmosphere. For the Tropical atmosphere, the spectrally integrated heating rate sensitivity profile shows a decrease of the heating rate at almost every level except for the first levels near the surface. The decrease is especially important in the $800 - 950 \ \mathrm{hPa}$ range. The panel (a2)

informs us about the spectral distribution of this decrease. On both the pure rotational and rotational-vibrational transition bands, where water vapor absorption is important, we observe a heating rate increase at the level of minimum heating rate (maximum cooling rate) and a heating rate decrease of the same magnitude above the level of minimum heating rate. This behavior corresponds to an elevation of the altitude of the minimum heating rate. The decrease of heating rate in the ozone absorption band ($980 - 1080 \ \mathrm{cm}^{-1}$ range) in the upper troposphere and the stratosphere illustrates the competition between

water vapor absorption and ozone absorption in this region of the longwave spectrum. Compared to the Tropical atmosphere, the impact of a 1% increase of water vapor concentration on the relatively dry Polar 2 atmosphere is less important. As represented in the panels (e2) and (f2) of Figure 7, the vertical heating rate sensitivity is negative for all levels. In the troposphere, the decrease is limited to $-0.07 \ \mathrm{K.day}^{-1}$. A noticeable decrease reaching $-0.2 \ \mathrm{K.day}^{-1}$ is present in the upper atmosphere with the same amplitude as on the Tropical atmosphere. Visible on the panel (d2), this decrease is caused by the combination of both

contribution of the pure rotational and rotational-vibrational transition bands of water vapor adding up at levels between $0.1$ and $1 \ \mathrm{hPa}$ while they compensate at higher levels. Here, we also notice the elevation of the altitude of the minimum heating rate in the troposphere.

Panels (a3) to (f3) of Figure 7 present the vertical heating rate sensitivity to a doubling in carbon dioxide concentration. As it can be seen on the panel (b3) and (e3), the heating rate profile is slightly affected in the troposphere and highly affected in the

stratosphere. In the Tropical case, we notice a small increase of the heating rate in the lower troposphere until $400 \ \mathrm{hPa}$. Above this level, the heating rate stays constant until the tropopause. In the Polar 2 atmosphere, we observe the same tropospheric effect with a diminished amplitude. The heating rate profile stops increasing at $400 \ \mathrm{hPa}$. In the stratosphere and above, Tropical and Polar 2 atmospheres show a similar sensitivity to the doubling of $CO_2$ concentration. In this pressure range, we observe an important decrease of the heating rate. At the TOA, the cooling to space is more intense in the Tropical atmosphere than in

the polar atmosphere.

The analysis of the sensitivity spectrum shows that the main impact of the doubling of $CO_2$ concentration on the heating rate is located in the $\nu_2$ absorption band. In band wings, we notice an increase of the heating rate in the lower troposphere and





a small decrease in the upper troposphere, followed by an important heating rate decrease in the stratosphere and above. In the band core however, no change occurs under the tropopause. We notice a small increase in the lower stratosphere followed

by a very high decrease in the upper stratosphere. In both atmospheres, we notice a small sensitivity in the stratosphere and mesosphere in the $2200 - 2400 \ \mathrm{cm}^{-1}$ range.

## 5    Conclusions

Taking advantage of its pre-calculated optical depth look-up table, the fast and accurate radiative transfer model 4A/OP calculates the radiance spectra for a user defined layered atmospheric model. In an effort to enhance the capabilities offered by

4A/OP, we have developed a module called 4A-Flux that computes the hemispheric radiative flux profiles, the net flux profile and the heating rate profile in clear-sky conditions under the assumption of local thermodynamic equilibrium, plane-parallel atmosphere and specular reflection on the surface. When the only required output is the OLR, it is possible to set an option that substantially decreases computation time. The angular integration is performed using the exponential integral functions $E_n$. The linear variation of the sublayer Planck function to the optical depth has been implemented to better represent the emission

of layers with a high optical depth.

With its 4A-Flux module, 4A/OP has contributed to RFMIP-IRF, providing the hemispheric radiative flux profiles. The hemispheric radiative flux profiles and OLR calculated by 4A/OP have been compared to the outputs of other contributing state-of-the-art radiative transfer models showing a good agreement between 4A/OP and the other models in terms of hemispheric flux profiles, the difference to the mean of all models being always lower than $0.49 \ \mathrm{W.m}^{-2}$ and almost always included in the

standard deviation of all models. In terms of OLR, the mean difference between 4A/OP and other models is $-0.148 \ \mathrm{W.m}^{-2}$ and the mean standard deviation is $0.218 \ \mathrm{W.m}^{-2}$, also showing a good agreement between 4A/OP and other models. We have also shown that 4A/OP outputs are especially close to LBLRTM.

Using 4A/OP models with 4A-Flux module, we have computed the sensitivities of the OLR and vertical heating rate to several perturbations of atmospheric and surface parameters. Our study confirms the typical OLR sensitivities to thermodynamic

and composition variables. We have also seen that the increase of the water vapor concentration leads to an elevation of the altitude of the minimum monochromatic heating rate. The spectral resolution of $5 \cdot 10^{-4} \ \mathrm{cm}^{-1}$ offered by 4A/OP allows very fine spectral sensitivity studies such as the effects of minor trace gases (e.g. HFC134a, HCFC22 and CFC113) on the OLR and vertical heating rate.

The development the 4A-Flux module and its implementation into 4A/OP radiative transfer code offers multiple possibili-

ties of studies on the clear-sky radiative fluxes and vertical heating rate at a very high spectral resolution and on any arbitrary atmospheric and surface description. It has already contributed to the RFMIP-IRF — along with several other reference radiative transfer models — to serve as benchmark models to characterize the accuracy of the parameterization of the IRF used in climate models. The very good agreement among these reference models while they differ in numerical methods, coding and spectroscopic databases are important factors in the confidence that we can have in their results. To go further and improve our

understanding of the remaining discrepancies between them, we plan to compare 4A/OP outputs using either the GEISA or the





HITRAN spectroscopic databases and thus be able to identify the differences in results and discriminate between those that are due to the spectroscopic properties of gases and those that are due to the radiative transfer modelling.

**Appendix A:  Complementary information on the evaluation of 4A-Flux over the RFMIP database**

This appendix presents further information on the differences between 4A-Flux and the other benchmark models of RFMIP-
IRF. Table A1 lists the 18 experiments used in RFMIP, more detailed information is provided in the protocol paper of RFMIP (Pincus et al., 2016). Figure A1 presents the detailed comparison of global-mean OLR (weighted average of the 100 atmospheric situations) between 4A/OP and the five other benchmark models of RFMIP-IRF. Here, experiments, forcing variants and physics variants are not averaged allowing a thorough comparison. Conditional errors that can possibly be hidden in the averaging process are here clearly visible. The results displayed in Table 3 are simply the averaged values over the experiment
indices presented in this figure.

As discussed in the main discussion, 4A/OP model is especially close the results of LBLRTM for the three forcing indices. We notice here that in terms of OLR, the three models GFDL-GRTCODE, LBLRTM and 4A/OP are close together. The distance between 4A/OP OLR is higher for the forcing variant f1 and experiments 4 and 17 that correspond to "future" gas concentrations. The OLR computed with 4A/OP is significantly higher due to the limited gas list that has been accounted for
in the simulation (16 out of the 43 specified in RFMIP database). These minor gases have small impact on the pre-industrial and present day radiative forcing but plays a greater part in the future radiative forcing. The concentration of some of these gases increase by orders of magnitude in the "future" scenario compared to the present day situation. The difference completely disappears for forcing variants f2 and f3 where the same ensemble of gases has been considered for all models. Gases that are neglected in the present day forcing can play an important part of the "future" forcing as their concentrations increase and thus
should be accounted for in such cases. If we exclude experiments 4 and 17 and ARTS physics variant p1 that does not include $CO_2$ line-mixing, the difference between 4A/OP and other models never exceeds $\pm 1$ $W.m^{-2}$.

*Code and data availability.*  The distributed version of 4A/OP radiative transfer model version 1.5 is available at https://4aop.aeris-data.fr/. This version of 4A/OP does not include 4A-Flux module version 1.0 yet. However, the development version of 4A/OP v1.5 including 4A-Flux v1.0 – described in this paper – is available on Zenodo platform (Tellier et al. (2021)). The spectroscopic database GEISA
(Gestion et Etude des Informations Spectroscopiques Atmosphériques: Management and Study of Atmospheric Spectroscopic Information) is available at https://geisa.aeris-data.fr/. The Thermodynamic Initial Guess Retrieval (TIGR) atmospheric database is available at https://ara.lmd.polytechnique.fr/index.php?page=tigr. All inputs and results for RFMIP experiment rad-irf are available on the Earth System Grid Federation at https://esgf-node.llnl.gov/search/cmip6/. Input data are referenced as Pincus (2019) and results data used for the calculations of sect. 3 are referenced as follows: 4A/OP v1.5 (Boucher et al., 2020), ARTS 2.3 (Brath, 2019), GRTCODE (Paynter et al., 2018a),
RFM-DISORT (Paynter et al., 2018b), HadGEM3-GC3.1 (Andrews, 2019) and LBLRTM 12.8 (Mlawer and Pernak, 2019).





**Table A1.** List of atmospheric conditions and gas concentrations defining the 18 experiments variants of RFMIP.

| Index | Atmospheric conditions | Gas concentrations |
|---|---|---|
| 1 | PD | PD |
| 2 | PD | PI GHG concentrations |
| 3 | PD | PI $4 \times CO_2$ |
| 4 | PD | "future" (RCP8.5 at 2100) |
| 5 | PD | PI $0.5 \times CO_2$ |
| 6 | PD | PI $2 \times CO_2$ |
| 7 | PD | PI $3 \times CO_2$ |
| 8 | PD | PI $8 \times CO_2$ |
| 9 | PD | PI $CO_2$ |
| 10 | PD | PI $CH_4$ |
| 11 | PD | PI $N_2O$ |
| 12 | PD | PI $O_3$ |
| 13 | PD | PI HFC (all HFC at zero) |
| 14 | PD +4K | PD |
| 15 | PD +4K | PD with increased relative humidity |
| 16 | PI | PI |
| 17 | "future" | "future" |
| 18 | PD | Last Glacial Maximum per PMIP |

Notes: for a complete presentation, please refer to the protocol paper of RFMIP (Pincus et al., 2016); PMIP stands for Paleoclimate Modelling Intercomparison Project (Kageyama et al., 2016).

*Author contributions.* Y. Tellier developed and implemented 4A-Flux module and handled the different applications with fruitful inputs from all authors. C. Crevoisier supervised the whole work and along with all authors, provided both scientific and technical expertise. J.-L. Dufresne also made the participation to RFMIP-IRF possible. Y. Tellier prepared the manuscript with contributions from all authors.

*Competing interests.* The authors declare that they have no conflict of interest.

*Acknowledgements.* Authors acknowledge *Thales Services Numériques* and the *Centre National d'Études Spatiales* for the financial support provided to this work. They also acknowledge Robert Pincus for the fruitful collaboration on RFMIP project along with all the teams that have contributed to RFMIP-IRF and publicly provided their model outputs used in this work. We acknowledge the World Climate Research Programme, which, through its Working Group on Coupled Modelling, coordinated and promoted CMIP6. We thank the climate modeling groups for producing and making available their model output, the Earth System Grid Federation (ESGF) for archiving the data and providing



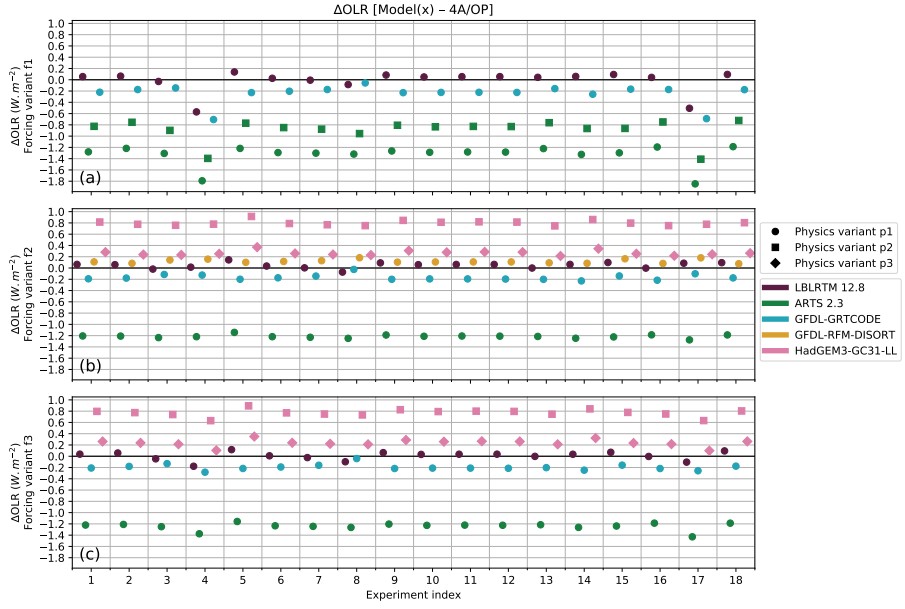

**Figure A1.** Means and standard deviations of OLR differences (model minus 4A/OP) in $\mathrm{W.m^{-2}}$, as a function of the experiment index. The five models compared to 4A/OP are represented by colors. The three forcing indices f1, f2 and f3 are represented on individual sub-figures respectively (a), (b) and (c). The different physics variants are differentiated with marker shapes.

access, and the multiple funding agencies who support CMIP6 and ESGF. Further acknowledgments are expressed to the *IPSL* and the support team of *Climserv* (the computation cluster of the IPSL) for the provision of both the computational power and the technical support that have been necessary to the completion of this work.





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
