# Peer review of "Computation of longwave radiative flux and vertical heating rate with 4A-Flux v1.0 as integral part of the radiative transfer code 4A/OP v1.5"

_Geoscientific Model Development, 2021_

## Referee Comment (RC2)

DETAILED COMMENTS for GMD Manuscript gmd-2021-325 "Computation of longwave radiative flux and vertical heating rate with 4A-Flux v1.0 as integral part of the radiative transfer code 4A/OP v1.5" by Yoann Tellier et al.

The paper describes the implementation and testing of a dedicated clear sky Outgoing Longwave radiation spectral module for the 4A/OP code. The paper outlines some of the math involved in changing the quadrature into exponential integrals. Comparison of the output against other models is performed. The spectral (quasi monochromatic) nature of the output allows the authors to present results that demonstrate how changes to trace gases change the OLR in expected spectral regions.

The nature of the paper (algorithm) means there is not much new scientific that can be presented in this paper. I find the paper well written, though I have some questions/comments (see below).

Based on what they have presented, I suggest to complement Section 4.2 on OLR sensitivity to perturbations of the profiles, can the authors also present a study of how OLR changes if they perturb the spectroscopy in their atlas? In other words, they have stated the flux computed by LBLRTM 12.8 (I believe using HITRAN 2012) is very close to what is computed by their model (using GEISA 2015). Can they also approach that from another avenue, namely can they add in the uncertainties of the (GEISA) database parameters such as line strengths, line shifts, air and self broadening etc, to their atlas, generate optical depths and then re-compute OLR to show how the spectral differences and total OLR change? The parameter changes could be done independently, and also randomly put together. These should be relatively straightforward to do, though it may be time consuming to generate new database(s).

Equation 6 (where the authors define  $E_n(x)$ ) could be augmented by some of the recurrence relations used to derive the final equations eg I assume  $\int^{x_0} E_2(x) dx = E_3(x_0)$ ?

Could the authors estimate or comment about expected differences in computed flux and heating rates using (a) constant, linear-in-tau or exponential-in-tau temperature variations and (b) how flux and heating rate calculations be impacted as you change the spectral resolution?

Below are some additional minor comments/suggestions

- 1. Traditionally the angular quadrature for eg upward flux is done by taking a stream of radiation leaving the surface at a particular angle, then doing the radiative transfer to the Top of Atmosphere at that angle; repeat for different angles and sum. The method outlined in this paper is a neat way of doing that, but they state it is for a plane parallel atmosphere. Can the authors comment how the output of their code would change if it were modified to account for spherical geometry (ie the atmosphere can be thought of as concentric shells around the earth)?
- 2. I believe Grant Petty's "Introduction to Atmospheric Radiation" also derives flux integrals in terms of exponential integrals of order n (but with the events of last couple years I seem to have misplaced my copy and can't verify this, sorry)
- 3. Abstract, line 13 : just use " and a standard deviation of"
- 4. Line 20-27 : The authors should be careful and differentiate between line-by-line models such as LBLRTM and 4A/OP, versus the rapid but less accurate flux calculators such as RRTM and ecRad, used in GCMs

- 5. Line 28 : please elaborate how "layers" in your model are defined for the atmosphere. For example Table 1 talks about 43 versus 61 vertical levels what is pressure at Top of Atmosphere for either scenario, and how would number of levels affect the accuracy. Maybe you could phrase it "satellite observations are modeled by dividing the atmosphere onto layers that are thin enough that the properties of any layer do not vary significantly eg Ttop-Tbottom is less than 5 K, water vapor mixing ratio changes by less than 1 percent across the layer, etc.
- 6. Line 30 : I assume by "radiation parametrization" the authors mean the optical depths used in the fast models?
- 7. Line 37: "with these six benchmark models among which 4A/OP LBL code" is an incomplete sentence.
- 8. Line 44 : "the radiance at the correct layer temperature, pressure levels ..."
- 9. Line 53 : does "contraction" mean "integration over angles"
- 10. Line 68 : Is the internal resolution of 4A/OP always 5e-4 cm-1 (ie from FIR to NIR)? Actually please first mention the OLR spectral range used by 4A/OP here (10-3250 cm-1)
- 11. Line 78 "The sublayer temperature in 4A-Flux varies linearly" ...
- 12. Line 104, beginning of Sec 2.2.1 "We focus on the downwelling flux in this section."
- 13. Line 106 "As we focus on the longwave only, the radiative contribution of deep space ( less than 3 K) to the downwelling flux ..."
- 14. Line 109 replace "injecting" with "inserting"
- 15. Line 126,127 : perhaps "assumption" rather than "hypothesis" would be a more appropriate word, in the last two lines of Page 5? and elsewhere in Page 6
- 16. Equation 16, perhaps more fully say  $H = \frac{g}{c_p} \frac{\partial F}{\partial p} = \frac{\partial T}{\partial t}$
- 17. Table 1 : I'm quite surprised calculations at all levels takes so much longer than OLR at TOA, which requires the calculation at all intermediate levels!
- 18. Page 11, caption for Fig 2 : rephrase "for every pair of model and physics variant"?
- Line 263 : I'm pretty sure LBLRTM12.8 uses HITRAN 2012, MT-CKD3.2 and CO2 linemixing by Lamoroux; can you explicitly state this and give similar spectroscopy references for 4A/OP (eg GEISA 2015 etc)
- 20. Figure 3 : variability of WV and O3 sub-panels could be better plotted using logs for the x-axis
- 21. line 339-344 : Please also see and refer to the new J. CLim paper "An Analytical Model for Spatially Varying Clear-Sky CO2 Forcing" Nadir Jeevanjee, Jacob T. Seeley, David Paynter, and Stephan Fueglistaler e: 09 Nov 2021 Print Publication: 01 Dec 2021 DOI: https://doi.org/10.1175/JCLI-D-19-0756.1 Page(s): 9463–9480
- 22. lines 380-400 : is there any particular physical significance to these "kinks" ... if so could you explain in a summary?
- 23. line 478 : This brings me back to the point I mentioned above ... the HITRAN and GEISA databases should be pretty similar, so perhaps you should consider the assessing the impact of uncertainties in the spectroscopic databases on OLR fluxes.

---

## Author Comment (AC1)

**Final author comments on behalf of all co-authors**

All co-authors would like to express gratitude to both referees for providing insightful comments on the submitted manuscript. We have worked to provide the best answers to all comments and questions and to bring necessary corrections and improvements to the revised manuscript.
Please find hereafter the author's answer (blue upright typed text) to each individual comments and questions raised the RCs (*black italic typed text*), on behalf of all co-authors.

**Author comments on RC1:**

*Referee comment on "Computation of longwave radiative flux and vertical heating rate with 4A-Flux v1.0 as integral part of the radiative transfer code 4A/OP v1.5" by Yoann Tellier et al., Geosci. Model Dev. Discuss., https://doi.org/10.5194/gmd-2021-325-RC1, 2021*

*The paper describes an extension of the radiative transfer model Automatized Atmospheric Absorption Atlas OPerational (4A/OP). The original version of the model enables line-by-line. simulations of radiances in the thermal spectral region. The extension 4A-Flux performs an angular integration of the radiances to obtain irradiances (fluxes) and heating rates, either vertically resolved or only at top of atmosphere (outgoing longwave radiation OLR). For the angular integration exponential integral functions are applied. The model has already articipated in the Radiative Forcing Model Intercomparison Project (RFMIP) and performed well. As an application, OLR and heating rate profiles are calculated using the Thermodynamic Initial Guess Retrieval (TIGR) atmospheric database as model input. The paper is generally well written and as a model description paper, it fits the scope of GMD, although it does not include substantial new concepts. I have some comments regarding the methodology which should be clarified before publication.*

*General comments:*

*- There are several publications about the calculation of line-by-line irradiances (fluxes), heating rates etc. in the thermal region (e.g. Buehler et al. 2006 and references therein) More scientific context including references to relevant publications should be included in the introduction.*

A short paragraph has been added to the introduction section in order to give more context on the line-by-line irradiance computation. More references have also been added to the introduction of the paper.

*- The most commonly used methods to compute irradiances (fluxes) and heating rates are two-stream methods, which provide accurate results (e.g., Zdunkowski, W., Trautmann, T., & Bott, A. (2007). Radiation in the Atmosphere: A Course in Theoretical Meteorology. Cambridge: Cambridge University Press.). Please explain and justify why you used the exponential integral functions to perform the angular integration.*

As a first implementation of the computation of irradiances in 4A/OP model, we have decided to implement the most exact solution for the angular integration. Under the minimal assumptions detailed in the paper, the formulation using exponential integral functions provides an exact solution to this integral. This precise solution will usefully serve as a reference for the future implementations of other types of angular integration that make use of approximations such as the two-stream approximation (a type of two-point quadrature) or other types of quadrature approaches that allow faster computation of irradiances.

*- The role of clouds and aerosols is not discussed. Can clouds/aerosols be handled by the 4A model? Can the model handle scattering at all? If not are there any plans to extend the model in this direction?*

4A/OP with 4A-Flux module is already capable of simulating radiative flux and heating rate with clouds that can be considered as black (emittance equal to 1) or grey bodies (emittance smaller than 1) in the longwave. This can be done by executing 4A/OP radiative transfer computations above and below the layers of such clouds. Concerning aerosols and other scattering particles, the implementation of scattering processes into the model is necessary. The 4A/OP model can already take scattering processes into account owing to its coupling to DISORT/LIDORT programs. It is consequently possible to simulate the radiances in the presence of scattering particles (e.g. Rayleigh scattering or aerosols). However, the radiative flux and heating rate currently implemented in 4A-Flux v1.0 is not capable of using the outputs of the scattering programs to simulate scattering processes yet. Such functionalities will be implemented in future versions of the code. However, as we are first focusing on atmospheres that are free from scattering particles and on the longwave spectral region, the Rayleigh scattering in $1/\lambda^4$ is completely negligible.

*- Is specular reflection a good approximation for natural surfaces? I suppose Lambertian surface reflection is a much more realistic assumption. Please justify why you included specular reflection.*

In this first implementation of 4A-Flux, we chose to implement a specular reflection which is not well suited for any type of surfaces. This choice may impact the computation of fluxes especially over deserts where the surface emittance can be quite low (around 0.7). However, on large regions of the longwave spectrum and over oceans, the emissivity is very high around 0,98 which implies that the reflected contribution only accounts for 2% of the upwelling radiance at the surface level. Then, the contribution of the surface to the upwelling radiance is less and less important as the altitude increases at wavenumber where the absorption is important (mainly outside of the IR window channel). Consequently, the TOA fluxes is less impacted by the assumption made about the surface reflection.

The implementation of an option to choose between a Lambertian and a specular surface is planned for the next version of 4A-Flux. This implementation is straightforward and even simplifies the computation of the reflected component of the upwelling flux. The only requirement is to replace in the third integral of Eq. 12 (referring hereafter to the revised paper's numbering) the factor that represents the reflected radiance at the surface of emittance $\epsilon$, in the direction $\mu$:

$$(1 - \epsilon)I^\downarrow(\sigma_{BOA}, \mu)$$

by the following Lambertian implementation:

$$\frac{(1 - \epsilon)F^\downarrow(\sigma_{BOA})}{\pi}$$

where $F^\downarrow(\sigma_{BOA})$ is the downwelling flux at the surface (computed by Eq. 11), and the $\pi$ factor is here to normalize the flux. By integration and using the exponential integral of order 3, such a replacement in Eq. 12 leads to the simple Lambertian reflected component that replaces Eq. 15 for every level $k$:

$$F^\uparrow_{refl}(\sigma_k) = \pi(1 - \epsilon)F^\downarrow_{refl}(\sigma_N)E_3(\Delta_{N,k}).$$

*Minor comments:*

*l. 53: "The spectral integration was performed with either a Gaussian quadrature or at a single angle under the diffuse approximation"- I assume you mean angular integration?*

Exactly, we mean angular integration and not spectral integration. This has been corrected in the revised paper.

*l. 107: "solution of the radiative transfer equation ..." -> here you may add that this solution is called Schwarzschild equation*

The mention of the Schwarzschild equation has been added to the revised paper.

*Eq. 16: Please define g and c_p*

The required definitions have been added to the revised paper.

*Table 1: I did not understand why the computational time for the vertical profiles is so much higher, could you explain this? For the calculation of OLR you need to step through all layers. Couldn't you save the results at all layer boundaries to get the vertical flux profiles?*

To calculate the OLR the only requirement is to add up the contributions of the emission of every layer $i$ that are transmitted to the TOA level. Therefore, the computational time is approximately proportional to the number of levels $N$. On top of the previous calculation of computational complexity $N$, the complete vertical profile requires the addition of the contributions of the emission of every layer $i$ transmitted to every levels $k$ above (resp. below) the layer $i$ for the upwelling (resp. downwelling) flux. Therefore, the computational time is proportional to $N(N + 1)/2$ and the computational complexity is $N^2$. The additional terms required for the complete vertical profile are not

required for the OLR calculation. This can be deduced from Eq. 11, 13 and 15 of the paper where it can be seen that the factors $E_3(\Delta_{i,j}) - E_3(\Delta_{i',j'})$ not only depend on the emission layers, but also on the level at which the flux are computed. This discussion about the computational time has been integrated to the paper to improve the readers' understanding.

*l. 387ff.: Which are the main processes responsible for the "kink"? Isn't this mainly due to absorption by the ozone layer? This would also explain why it appears at different altitudes for the different scenarios,*

As clearly demonstrated by Jeevanjee and Fueglistaler (2020), the upper-tropospheric kink in the heating rate profile originates from the distribution of absorption coefficients in the water vapor rotational band and is not related to the stratospheric ozone. More precisely, the authors show that, in the $H_2O$ rotational band, the occurrence of more weakly absorbing wavenumbers is relatively constant whereas the occurrence of more strongly absorbing wavenumbers sharply decline above a certain threshold corresponding to the kink observed on the heating rate profile. Being linked to the water vapor absorption, this also explain why the kink appears at different altitudes for the different scenarios. This explanation has been added to the paper in order to improve the interpretation of the heating rate profiles.

*Fig. 7: (c1) and (f1) do not show heating rate difference but heating rate. (c2) and (f2) show exactly the same as (b2) and (e2), the same scale on the x-axis.*

Figure 7 has been corrected and modified in accordance with the referee's recommendations. (c2) and (f2) have been adapted to zoom on the stratospheric variations as the sensitivity of the heating rate to $H_2O$ concentration is greater in the troposphere than in the stratosphere (mainly due to the relatively higher concentration of water vapor in the troposphere).

*References:*
*Buehler, SA, A von Engeln, E Brocard, VO John, T Kuhn, and P Eriksson (2006), Recent developments in the line-by-line modeling of outgoing longwave radiation, JQSRT, 98(3), 446-457*
*Zdunkowski, W., Trautmann, T., & Bott, A. (2007). Radiation in the Atmosphere: A Course in Theoretical Meteorology. Cambridge: Cambridge University Press*

References:
- Jeevanjee, N. and Fueglistaler, S.: Simple Spectral Models for Atmospheric Radiative Cooling, Journal of the Atmospheric Sciences, 77, 479–497, https://doi.org/10.1175/JAS-D-18-0347.1, 2020.

*DETAILED COMMENTS for GMD Manuscript gmd-2021-325 "Computation of longwave radiative flux and vertical heating rate with 4A-Flux v1.0 as integral part of the radiative transfer code 4A/OP v1.5" by Yoann Tellier et al.*

*The paper describes the implementation and testing of a dedicated clear sky Outgoing Longwave radiation spectral module for the 4A/OP code. The paper outlines some of the math involved in changing the quadrature into exponential integrals. Comparison of the output against other models is performed. The spectral (quasi monochromatic) nature of the output allows the authors to present results that demonstrate how changes to trace gases change the OLR in expected spectral regions.*

*The nature of the paper (algorithm) means there is not much new scientific that can be presented in this paper. I find the paper well written, though I have some questions/comments (see below).*

*Based on what they have presented, I suggest to complement Section 4.2 on OLR sensitivity to perturbations of the profiles, can the authors also present a study of how OLR changes if they perturb the spectroscopy in their atlas? In other words, they have stated the flux computed by LBLRTM 12.8 (I believe using HITRAN 2012) is very close to what is computed by their model (using GEISA 2015). Can they also approach that from another avenue, namely can they add in the uncertainties of the (GEISA) database parameters such as line strengths, line shifts, air and self-broadening etc, to their atlas, generate optical depths and then re-compute OLR to show how the spectral differences and total OLR change? The parameter changes could be done independently, and also randomly put together. These should be relatively straightforward to do, though it may be time consuming to generate new database(s).*

In this paper, we have decided to remain focused on the description of 4A-Flux module and some first applications of the computation of the radiative flux and heating rate by performing a straightforward sensitivity study. The simulations are based on GEISA 2015 and compares very well with LBLRTM that is using HITRAN 2012 (Pincus et al., 2020). This comparison indeed calls for a study about the influence of the spectroscopic parameters on the spectral or total OLR and heating rates. Unlike the influence on the spectral quantities that is expected to be important, the impact on the spectrally integrated quantities is expected to be small. Indeed, perturbations on the spectroscopic parameters are redistributing the energy only locally (energy conservation).

Even if such a study is conceptually straightforward, its application requires to generate a fair number (for statistical significance) of atlases affected by independent or random (bias and standard deviation) perturbations on many spectroscopic parameters such as line strengths, line shifts, air and self-broadening as suggested by the reviewer. The generation of these atlases requires a complete radiative transfer model (STRANSAC) that needs a lot more computational time than the fast model 4A/OP does.

Furthermore, the question of the values of the perturbations to be applied cannot be overlooked. Being both time consuming and of great interest, we have considered that such a study would deserve an independent paper and will be performed with the following versions of 4A/OP model.

*Equation 6 (where the authors define $E_n(x)$) could be augmented by some of the recurrence relations used to derive the final equations eg I assume $\int^{x_0} E_2(x)dx = E_3(x_0)$?*

The properties of the exponential integral functions have been included in the paper (recurrence relations and value at the origin).

*Could the authors estimate or comment about expected differences in computed flux and heating rates using (a) constant, linear-in-tau or exponential-in-tau temperature variations and (b) how flux and heating rate calculations be impacted as you change the spectral resolution?*

(a) In the paper we have briefly introduced two assumptions about the sublayer variation of the Planck function $B(T(\sigma))$. The first one $B(T(\sigma))$ is assumed constant in the layer and evaluated at the mean temperature of the adjacent levels $\bar{T} = (T_{up} + T_{down})/2)$, the second approach assumes the Planck function (not the temperature) to be linear as a function of the layer optical depth $\sigma$. When the layer optical depth is small, the two assumptions are equivalent as it can be calculated from Eq. 10. However, when the layer optical depth is high, the layer's emission tends to the Planck's law at temperature of the lower (resp. upper) level directly adjacent to the layer for the downwelling (resp. upwelling) radiation. The comparison performed by Ridgeway et al. (1991) shows the constant $B(T(\sigma))$ approximation introduces a systematic cold (resp. warm) bias in estimated downwelling (resp. upwelling) fluxes. It also shows small differences between the longwave integrated heating rates computed under the linear-in-tau approximation and the constant $B(T(\sigma))$ approximation. However, one can expect a more important bias on the monochromatic heating rates where the optical depth is high. This could be verified by systematically comparing heating rates spectra computed under each assumption. Furthermore, the thicker the layers are (low number of vertical layers), the higher the bias caused by the constant $B(T(\sigma))$ approximation is. Thus, the linear-in-tau approximation allows to preserve accuracy even on coarser vertical resolution atmospheres (faster computations). The linear-in-tau approximation previously used is currently being completed by the constant approximation for the next version of 4A-Flux in order to perform thorough comparisons.
The exponential-in-tau approximation has not been studied here and is not familiar to the authors of the paper.

(b) All the computations in the paper (fluxes, heating rates) have been performed at the spectral resolution of the Atlas ($5\times10^{-4}$ cm$^{-1}$) to preserve the highest accuracy. The spectral integrations are performed *a posteriori* and do not affect the precision on the outputs.

*Below are some additional minor comments/suggestions*

*Traditionally the angular quadrature for eg upward flux is done by taking a stream of radiation leaving the surface at a particular angle, then doing the radiative transfer to the Top of Atmosphere at that angle; repeat for different angles and sum. The method outlined in this paper is a neat way of doing that, but they state it is for a plane parallel atmosphere. Can the authors comment how the output of their code would change if it were modified to account for spherical geometry (ie the atmosphere can be thought of as concentric shells around the earth)?*

The formalism using the exponential integrals $E_n(x)$ can only be used under the assumption of plane parallel geometry. Indeed, the mathematical definition of $E_n(x)$ assumes that $\mu$ does not depend on the atmospheric level which cannot be true under the assumption of spherical geometry.
The spherical geometry is not used in version v1.0 of 4A-Flux as we have chosen to use the exponential integral function. However, it has been implemented in 4A/OP and commonly used for the computation of the radiances.

A previous (unpublished) prototype version of the 4A-Flux could handle the spherical geometry performing an angular quadrature by computing 30 radiance streams equally spaced over the half-hemisphere (azimuthal symmetry) and computing the flux using the trapezoidal rule. Note that more accurate quadrature methods can be used such as the Gauss-Legendre quadrature to evaluate the integral without decreasing accuracy and using less streams (angles) leading to smaller computation time.
Such implementations using quadrature are compatible with the spherical geometry. The radiative flux can be computed by replacing slant optical depth $\sigma/\mu_0$, where $\sigma$ is the optical depth and $\mu_0$ is the cosine of the angle of the stream at the surface, by $\sigma Ch(r, \mu_0)$ where $r$ is the altitude of the considered level in the spherical geometry and $Ch(r, \mu_0)$ is the air-mass factor also called the Chapman function (see appendix M of Stamnes et al., 2017).
The output of the 4A-Flux would probably be affected by the implementation of the spherical geometry even if the actual impact has not been evaluated yet. This functionality is part of the improvements that are planned to be implemented in a future version of 4A-Flux.

*I believe Grant Petty's "Introduction to Atmospheric Radiation" also derives flux integrals in terms of exponential integrals of order n (but with the events of last couple years I seem to have misplaced my copy and can't verify this, sorry)*

Thank you very much for this information, if we manage to access this resource we would carefully read it. Note that the development of radiative flux in terms of exponential integral functions is also presented in Stamnes et al. (2017).

*Abstract, line 13 : just use " and a standard deviation of"*

The paper has been modified to follow this recommendation.

*Line 20-27 : The authors should be careful and differentiate between line-by-line models such as LBLRTM and 4A/OP, versus the rapid but less accurate flux calculators such as RRTM and ecRad, used in GCMs*

Following the referee's suggestion, the distinction has between line-by-line and rapid models has been made clearer in the first paragraph of the paper. More references have also been added to the paragraph.

*Line 28 : please elaborate how "layers" in your model are defined for the atmosphere. For example Table 1 talks about 43 versus 61 vertical levels – what is pressure at Top of Atmosphere for either scenario, and how would number of levels affect the accuracy. Maybe you could phrase it "satellite observations are modeled by dividing the atmosphere onto layers that are thin enough that the properties of any layer do not vary significantly eg Ttop-Tbottom is less than 5 K, water vapor mixing ratio changes by less than 1 percent across the layer, etc.*

The sampling of the vertical profiles of pressure and temperature used here have been set during the development of 4A/OP fast model in the context of the generation of the Atlas that must preserve high precision and fast computation. The 43 levels have been set to remain, for every wavenumber, in the linearity range that allow such precision and rapid interpolations performed by 4A/OP. In order to minimize the errors in the computation of radiative flux, we have set the TOA layer to be very high in the atmosphere, at $2.6 \times 10^{-3}$ hPa in the TIGR database. In the RFMIP database the vertical profile is divided into 61 levels and the value of the TOA pressure is $1.0 \times 10^{-2}$ hPa. Both pressure levels correspond to an altitude ranging between 80 and 100 km depending on the airmass type (tropical or polar).

*Line 30 : I assume by "radiation parametrization" the authors mean the optical depths used in the fast models?*

By "radiation parameterization", the authors mean fast models used in GCMs that parameterize the radiative transfer. This has been clarified in the paper. Such models make use of a set of parameters which under specific assumptions allows the very fast computation of radiative transfer at the cost of reduced accuracy.

*Line 37 : "with these six benchmark models among which 4A/OP LBL code" is an incomplete sentence.*

The sentence has been completed in the paper.

*Line 44 : "the radiance at the correct layer temperature, pressure levels ..."*

The sentence has been corrected in the paper.

*Line 53 : does "contraction" mean "integration over angles"*

In this sentence, the first occurrence of the words "spectral integration" has been corrected and replaced by "angular integration". By "contraction" we actually mean

"spectral integration". Here, we wanted to explain that in order to perform the angular quadrature in a reasonable amount of time, the method that used to be implemented performed a spectral integration prior to the angular quadrature. This process used to reduce the accuracy of the radiative flux computation. This is no longer the case in 4A-flux.

*Line 68 : Is the internal resolution of 4A/OP always 5e-4 cm-1 (ie from FIR to NIR)? Actually please first mention the OLR spectral range used by 4A/OP here (10-3250 cm-1)*

The Atlas upon which 4A/OP rapidly computes the radiative transfer can be generated from spectroscopic databases at any desired spectral resolution. In this paper, the resolution of the Atlas has been set at $5.0 \times 10^{-4}$ cm$^{-1}$ for it corresponds to the full-width at half maximum of the thinnest absorption lines in the thermal infrared (TIR) region. The absorption lines becoming larger and larger as the wavelength shortens, this half-width is also sufficient to model the short-wave infrared and near-infrared. However, at longer wavelength, for instance in the far infrared (FIR) region, the half-width of the lines could be significantly narrower than $5.0 \times 10^{-4}$ cm$^{-1}$. In the context of development of FORUM, the ESA ninth Earth Explorer mission (Palchetti et al., 2020), LMD has planned activities among which an ongoing study to determine the required spectral resolution of the Atlas in the FIR region of the spectrum

*Line 78 "The sublayer temperature in 4A-Flux varies linearly" ...*

With this implementation, we take account of a certain variation of the temperature at the sublayer scale. However, the variation of temperature inside layers has not been taken linear. It is the Planck's function value that varies linearly as the function of the optical depth between its value at the temperature at levels (interfaces of the layers) and the average temperature of the layers. This assumption lead to a better representation of the actual temperature at which the layer is emitting radiation.

*Line 104, beginning of Sec 2.2.1 "We focus on the downwelling flux in this section."*

The paper has been modified in accordance with the referee's recommendation.

*Line 106 "As we focus on the longwave only, the radiative contribution of deep space (less than 3 K) to the downwelling flux ..."*

The precision has been added to the paper.

*Line 109 replace "injecting" with "inserting"*

The paper has been corrected accordingly.

*Line 126,127 : perhaps "assumption" rather than "hypothesis" would be a more appropriate word, in the last two lines of Page 5? and elsewhere in Page 6*

The paper has been modified in accordance with the referee's recommendation.

*Equation 16, perhaps more fully say $H = \frac{g}{c_p}\frac{\partial F}{\partial p} = \frac{\partial T}{\partial t}$*

The paper has been modified in accordance with the referee's recommendation.

*Table 1 : I'm quite surprised calculations at all levels takes so much longer than OLR at TOA, which requires the calculation at all intermediate levels!*

As we answered to the first referee comments, in order to calculate the OLR the only requirement is to add up the contributions of the emission of every layer $i$ that are transmitted to the TOA level. Therefore, the computational time is approximately proportional to the number of levels $N$. On top of the previous computation of computational complexity $N$, the complete vertical profile requires the addition of the contributions of the emission of every layer $i$ transmitted to every level $k$ above layer $i$ (resp. below) for the upwelling (resp. downwelling) flux. Therefore, the computational time is proportional to $N(N+1)/2$ and the computational complexity is $N^2$. The additional terms required for the complete vertical profile are not required for the OLR calculation. This can be deduced from Eq. 11, 13 and 15 of the manuscript where it can be seen that the factors $E_3(\Delta_{i,j}) - E_3(\Delta_{i',j'})$ not only depend on the emission layers, but also on the level at which the flux are computed. This discussion about the computational time has been integrated to the paper to improve the reader's understanding.

*Page 11, caption for Fig 2 : rephrase "for every pair of model and physics variant"?*

The paper has been modified in accordance with the referee's recommendation.

*Line 263 : I'm pretty sure LBLRTM12.8 uses HITRAN 2012, MT-CKD3.2 and CO2 linemixing by Lamoroux; can you explicitly state this and give similar spectroscopy references for 4A/OP (eg GEISA 2015 etc)*

For our study the spectroscopic database used by this version of 4A/OP is GEISA-2015 (Jacquinet-Husson et al., 2016). Like LBLRTM, the continuum used is the MT_CKD 3.2 continuum (Mlawer et al., 2012) and the $CO_2$ line-mixing, we used is based on the HITRAN-2012 parameters (Lamouroux et al, 2015). This fact is now mentioned in the revised paper. Note that all the required tools to compute the $CO_2$ line-mixing based on GEISA are now available at LMD and will be included in the next version of 4A/OP. This will be performed in the frame of the 2022 edition of GEISA database.

*Figure 3 : variability of WV and O3 sub-panels could be better plotted using logs for the x-axis*

Figure 3 of the paper has been modified in accordance with the referee's recommendation.

*line 339-344 : Please also see and refer to the new J. CLim paper "An Analytical Model for Spatially Varying Clear-Sky CO2 Forcing" Nadir Jeevanjee, Jacob T. Seeley, David Paynter, and Stephan Fueglistaler e: 09 Nov 2021 Print Publication: 01 Dec 2021 DOI: https://doi.org/10.1175/JCLI-D-19-0756.1 Page(s): 9463–9480*

We are thankful to the referee for the suggestion of this very interesting paper. It provides a great insight into the underlying cause of the results we presented in our paper. It explains that in the absence of $H_2O$, the $CO_2$ forcing can be considered as a swap of surface emission in band wings for the stratospheric emission in the band center, and that in the presence of $H_2O$, the surface emission in band wings is being replaced by the emission of a colder atmosphere. It echoes the observed forcing patterns close to the $CO_2$ band wings in Fig. 5 of our paper where in the tropical atmosphere (high RH) we observe a high sensitivity shared between $H_2O$ and $CO_2$ whereas in the polar 2 atmosphere (low RH) the high sensitivity is shared between surface temperature and $CO_2$. This paper is now referred to in the revised version of the manuscript providing a valuable conceptual tool for the interpretation of our results.

*lines 380-400 : is there any particular physical significance to these "kinks" ... if so could you explain in a summary?*

The physical significance of the "kinks" has been explained as a response to referee 1 and clearly demonstrated by Jeevanjee an Fueglistaler (2020). The upper-tropospheric kink in the heating rate profile originates from the distribution of absorption coefficients in the water vapor rotational band and is not related to the stratospheric ozone. More precisely, the authors show that, in the $H_2O$ rotational band, the occurrence of more weakly absorbing wavenumbers is relatively constant whereas the occurrence of more strongly absorbing wavenumbers sharply decline above a certain threshold corresponding to the kink observed on the heating rate profile. Being linked to the water vapor absorption, this also explain why the kink appears at different altitudes for the different scenarios. This explanation has been added to the paper in order to improve the interpretation of the heating rate profiles.

*line 478 : This brings me back to the point I mentioned above ... the HITRAN and GEISA databases should be pretty similar, so perhaps you should consider the assessing the impact of uncertainties in the spectroscopic databases on OLR fluxes.*

The impact of replacing switching GEISA and HITRAN databases for the generation of the Atlas to compute and compare the results in terms of OLR and heating rate is currently being investigated as a first step to the broader study of the influence of the spectroscopic parameter on the radiative quantities that has been briefly described in the answer to the first question of the referee. 4A-Flux module, as part of 4A/OP model now enable us to perform such studies.

References:

- Delahaye, T., Armante, R., Scott, N. A., Jacquinet-Husson, N., Chédin, A., Crépeau, L., Crevoisier, C., Douet, V., Perrin, A., Barbe, A., Boudon, V., Campargue, A., Coudert, L. H., Ebert, V., Flaud, J.-M., Gamache, R. R., Jacquemart, D., Jolly, A., Kwabia Tchana, F., Kyuberis, A., Li, G., Lyulin, O.

M., Manceron, L., Mikhailenko, S., Moazzen-Ahmadi, N., Müller, H. S. P., Naumenko, O. V., Nikitin, A., Perevalov, V. I., Richard, C., Starikova, E., Tashkun, S. A., Tyuterev, V. G., Vander Auwera, J., Vispoel, B., Yachmenev, A., and Yurchenko, S.: The 2020 edition of the GEISA spectroscopic database, Journal of Molecular Spectroscopy, 380, https://doi.org/https://doi.org/10.1016/j.jms.2021, 2021.

- Jeevanjee, N. and Fueglistaler, S.: Simple Spectral Models for Atmospheric Radiative Cooling, Journal of the Atmospheric Sciences, 77, 479–497, https://doi.org/10.1175/JAS-D-18-0347.1, 2020.
- Palchetti, L., Brindley, H., Bantges, R., Buehler, S. A., Camy-Peyret, C., Carli, B., Cortesi, U., Bianco, S. D., Natale, G. D., Dinelli, B. M., Feldman, D., Huang, X. L., C.-Labonnote, L., Libois, Q., Maestri, T., Mlynczak, M. G., Murray, J. E., Oetjen, H., Ridolfi, M., Riese, M., Russell, J., Saunders, R., and Serio, C.: FORUM: Unique Far-Infrared Satellite Observations to Better Understand How Earth Radiates Energy to Space, Bulletin of the American Meteorological Society, 101, E2030–E2046, https://doi.org/10.1175/BAMS-D-19-0322.1, 2020.
- Pincus, R., Buehler, S. A., Brath, M., Crevoisier, C., Jamil, O., Evans, K. F., Manners, J., Menzel, R. L., Mlawer, E. J., Paynter, D., Pernak, R. L., and Tellier, Y.: Benchmark Calculations of Radiative Forcing by Greenhouse Gases, Journal of Geophysical Research: Atmospheres, 125, https://doi.org/10.1029/2020jd033483, 2020.
- Ridgway, W. L., Harshvardhan, and Arking, A.: Computation of atmospheric cooling rates by exact and approximate methods, Journal of Geophysical Research: Atmospheres, 96(D5), 8969–8984, https://doi.org/10.1029/90JD01858, 1991.
- Stamnes, K., Thomas, G. E., and Stamnes, J. J.: Radiative Transfer in the Atmosphere and Ocean, Cambridge University Press, 2 edn., https://doi.org/10.1017/9781316148549, 2017.